# Techno-Economic Analysis of State-of-the-Art Carbon Capture Technologies and Their Applications: Scient Metric Review

Raghad Adam [1] and Bertug Ozarisoy [1,2,*]

1   Sustainable Environment and Energy Systems (SEES) Graduate Program, Middle East Technical University, Northern Cyprus Campus, Kalkanli, Guzelyurt 99738, Turkey; r.tahadam@gmail.com
2   School of the Built Environment and Architecture, London South Bank University, 103 Borough Road, London SE1 0AA, UK
*   Correspondence: ozarisoybertug@gmail.com

**Abstract:** Carbon dioxide ($CO_2$) emissions are a serious hazard to human life and the ecosystem. This is the reason that many measures have been put in place by the International Energy Agency (IEA) to reduce the anthropogenic-derived $CO_2$ concentration in the atmosphere. Today, the potential of renewable energy sources has led to an increased interest in investment in carbon capture and storage technologies worldwide. The aim of this paper is to investigate state-of-the-art carbon capture and storage (CCS) technologies and their derivations for the identification of effective methods during the implementation of evidence-based energy policies. To this extent, this study reviews the current methods in three concepts: post-combustion; pre-combustion; and oxy-fuel combustion processes. The objective of this study is to explore the knowledge gap in recent carbon capture methods and provide a comparison between the most influential methods with high potential to aid in carbon capture. The study presents the importance of using all available technologies during the post-combustion process. To accomplish this, an ontological approach was adopted to analyze the feasibility of the CCS technologies available on the market. The study findings demonstrate that priority should be given to the applicability of certain methods for both industrial and domestic applications. On the contrary, the study also suggests that using the post-combustion method has the greatest potential, whereas other studies recommend the efficiency of the oxy-fuel process. Furthermore, the study findings also highlight the importance of using life cycle assessment (LCA) methods for the implementation of carbon capture technologies in buildings. This study contributes to the energy policy design related to carbon capture technologies in buildings.

**Keywords:** adsorption; carbon capture methods; incineration; oxy-fuel combustion; post-combustion technologies

## 1. Introduction

Climate change has been caused by a dramatic increase in greenhouse gases (GHG) [1]. This global problem has led to risks both to human health and the built environment worldwide. Warming climate conditions have caused coastal erosion and longer growing seasons in the agriculture industry. Additionally, a rise in global temperatures could result in melting glaciers and ice sheets and a rapid rise in sea levels. This phenomenon has caused changes in global weather and climate patterns. This is the reason that humankind has been experiencing more frequent floods, droughts, typhoons, and cyclones in the last three decades. In 2022, the Paris Agreement was put in place at the United Nations Climate Change Conference (COP21). This global summit was aimed at limiting global warming to below 2 °C, taking the baseline as pre-industrial temperature levels. This has led to efforts to limit the global temperature rise to 1.5 °C above pre-industrial levels [2]. To achieve this, the Paris Agreement requires all United Nations countries to set strict $CO_2$ emissions reduction targets, which are directly related to climate change and mitigation strategies.

In response to the findings of the IPCC's Special Report on Global Warming, the United Nations has called for an urgent plan to reduce $CO_2$ emissions and limit global temperatures to well below 1.5 °C [3]. To achieve this target, the IPCC report has recommended as a mandatory task for all UN countries to reduce $CO_2$ emissions by 2030 and reach the goals of 'net-zero' emissions by the mid-century. This would require a rapid behavioral change in consumers' habits of energy use [3]. This has also been put into effect for producing goods, planning effective land management, and avoiding deforestation. To achieve the IPCC's goals, increasing the use of renewable energy sources should be promoted, including solar and wind power systems, and the deployment of cost-effective emission technologies that aid in removing $CO_2$ emissions from the atmosphere.

In 2018, the summary for policymakers of the IPCC Special Report indicated that MAGICC and FAIR models are correlated with geophysical data gathered from simulations. The study found that there is a high degree of variation within a specific category and between the models [4]. To this extent, many forecasting scenarios propose a temporary increase in global temperatures. This finding has been noted in previous studies on stringent mitigation efforts to reduce $CO_2$ emissions. Many scientists found that there is a very limited methodological framework, which has been considered the database approach to keeping future global warming below 1.5 °C throughout the 21st century [5]. However, most studies highlighted that there is an inconsistent pathway in the global databases in order to limit the global temperature rise to 1.5 °C [6,7]. This set threshold limit is expected to reach the benchmark criteria around the mid-century before lowering global temperatures to below pre-industrial records by 2100 [8]. However, it is difficult to distinguish the exact temperature characteristics of the different pathway categories due to various uncertainties and model dependencies in the simulation models. The extant literature studies stress that there is a possibility of reaching the exceedance limit of the average rise of 1.5 °C in global temperature by the mid-century because of the lack of data available on the assessment of the reliability of the weather data time series [9].

In accordance with the Paris Agreement and the POST-2020 framework, numerous nations have come to a consensus to tackle the pressing issue of global warming and its adverse environmental consequences. To mitigate the risks posed by climate change, these countries have established distinct objectives aimed at reducing greenhouse gas emissions. As an illustration, the European Union (EU) has committed to reducing its $CO_2$ emissions by 40% by the year 2030. Simultaneously, the United States (US) has set its target for a reduction of 26–28% by the year 2025. A comprehensive breakdown of these targets can be found in Table 1.

**Table 1.** Representative countries' $CO_2$ emissions reduction targets.

| Country | $CO_2$ Emissions Reduction Target | Target Year |
|---|---|---|
| The EU | 40% | 2030 |
| The USA | 26–28% | 2025 |
| China | 60–65% | 2030 |
| South Korea | 37% | 2030 |
| Japan | 26% | 2030 |
| Malaysia | 45% | 2030 |

Data source [10].

These targets represent significant efforts on the part of the international community to combat climate change and preserve our planet's delicate ecological balance. By actively pursuing these goals, nations are taking crucial steps towards a more sustainable and environmentally conscious future [10]. This study aims to address several gaps in the literature related to carbon capture and sequestration (CCS) in order to outline a state-of-the-art techno-economic analysis in the building sector. The study also seeks to provide an in-depth overview of the current technologies that could help mitigate climate change impacts by 2030 in line with the Paris Agreement regulations. Having reviewed recent research articles published in this research area, the main objective of this present study is to

research modern carbon capture and storage technologies and evaluate their technological specifications as well as their economic feasibility.

This study also aims to answer the question of whether CCS has the potential to mitigate $CO_2$ emissions as well as climate change effects in the built environment. To this extent, the significance of post- and oxy-fuel combustions is explored to determine a reliable assessment both in industrial and domestic applications. The novelty of the present study is to develop a cost-effective assessment criterion for CCS technologies by adopting a universal design approach in any type of building application, as shown in Figure 1. One of the main contributions to the knowledge is to develop economically viable CCS technologies to lower global temperatures and remove $CO_2$ concentrations from the atmosphere.

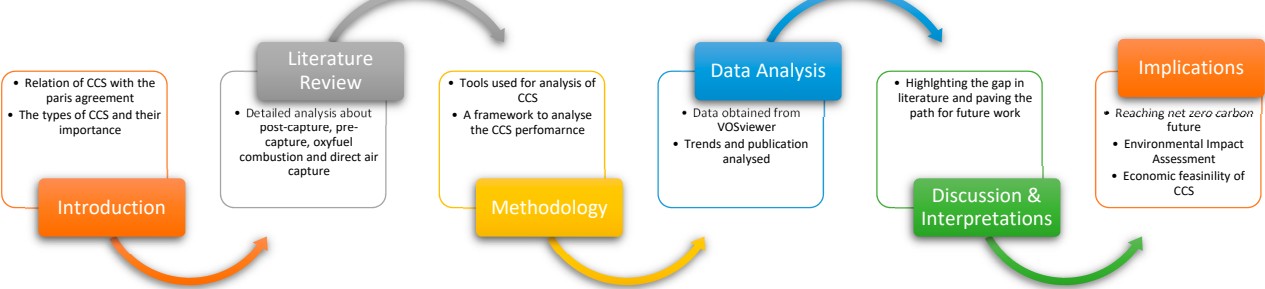

**Figure 1.** Conceptualization framework of the present study. Drawn by author.

The intention of the present study is to analyze the characteristics of oxy-fuel combustion to develop an effective optimization model for use in power generation plants. To accomplish this, a comprehensive methodological framework is produced to outline the policy design implications. The life cycle cost assessment approach was also explored to validate the adopted optimization models. Additionally, the findings were supported using the meta-analysis approach to gather research outputs from a systematic literature review. Policy design implications were outlined to demonstrate a roadmap for using carbon capture technologies and their applications. The systematic literature review presents a keyword analysis to determine a new method of design for the cost-effectiveness of carbon capture technologies. To fill the knowledge gap, a comprehensive research methodology framework was also outlined. A bibliometric review of all available methods was presented. Furthermore, the extant literature review analysis of state-of-the-art CCS technologies was investigated globally.

This paper set out to execute the state-of-the-art methodological framework considering the significance of CCS technologies in building design applications. The brief historical background is presented in Section 2. To this extent, the systematic literature review was undertaken by gathering original research papers, review articles, and policy documents worldwide. The conceptual framework is outlined to demonstrate the cost-effectiveness of implementing CCS technologies. Meta-analysis findings are presented in Section 4. The discussion is delineated to provide the applicability of three methods, namely, post-, pre-, and oxy-fuel combustion processes in Section 5. Conclusions are drawn to demonstrate the significance of the research outputs in Section 6.

## 2. Literature Review

### 2.1. Causes and Effects of Carbon Dioxide in the Atmosphere

The presence of carbon in the environment stems from various human activities, with the most significant concern being the release of excessive carbon dioxide ($CO_2$). One major source of $CO_2$ emissions is the combustion of fossil fuels, such as coal, oil, and natural gas, for energy production, transportation, and industrial processes. As these fuels burn, they release large quantities of carbon dioxide into the atmosphere, contributing significantly to the greenhouse effect and global warming. Another significant cause is deforestation,

which involves the clearing of forests for logging, agriculture expansion, or urbanization. Trees act as crucial carbon sinks, absorbing $CO_2$ during photosynthesis, but deforestation reduces their numbers, leading to increased carbon dioxide levels in the atmosphere.

Industrial activities also contribute to carbon emissions in the environment. Certain industrial processes, including cement production, chemical manufacturing, and other heavy industries, release substantial amounts of carbon dioxide as a byproduct. These emissions collectively add to the overall carbon footprint. Additionally, agricultural practices and land-use changes play a role in carbon release. Agriculture, through activities such as rice cultivation, livestock farming, and the use of synthetic fertilizers, releases methane $(CH_4)$ and nitrous oxide $(N_2O)$, which are potent greenhouse gases. Furthermore, land-use changes, like converting forests into agricultural land or urban areas, release stored carbon, further reducing the Earth's capacity to sequester carbon.

The harmful effects of excessive carbon in the environment have profound consequences for both society and the natural world. One of the most pressing issues is climate change, driven by the accumulation of greenhouse gases, particularly carbon dioxide, in the atmosphere. Climate change leads to rising global temperatures, resulting in more frequent and severe heatwaves, extreme weather events, and altered precipitation patterns. Such changes disrupt ecosystems, threaten agriculture, and impact human livelihoods and infrastructure. Moreover, climate change exacerbates health problems, leading to an increased prevalence of heat-related illnesses, worsened air quality and respiratory issues, and the spread of infectious diseases as disease vectors expand their range.

The harmful effects of carbon in the environment also extend to economic disruptions. Climate change-related events, such as hurricanes, floods, and droughts, can cause significant economic losses, disrupt supply chains, and impact industries, leading to financial instability and social upheaval. Additionally, the environment suffers from the consequences of excessive carbon emissions. For instance, the increased absorption of atmospheric carbon dioxide by the oceans leads to ocean acidification, harming marine life, especially organisms with calcium carbonate shells or skeletons, such as corals, mollusks, and some planktonic species. Furthermore, global warming caused by carbon emissions accelerates the melting of glaciers and ice sheets, contributing to rising sea levels. This poses a threat to low-lying coastal areas, leading to coastal erosion, the loss of habitats, and the displacement of human populations. Additionally, climate change disrupts ecosystems, making it challenging for many species to adapt or survive, resulting in shifts in the distribution of plants and animals and increasing the risk of extinction for numerous species.

Addressing the causes and effects of carbon in the environment is crucial for mitigating climate change and safeguarding both society and the planet. Transitioning to renewable energy sources, such as solar, wind, and hydroelectric power, can significantly reduce carbon emissions from energy production. Promoting sustainable land-use practices, protecting forests, and encouraging reforestation efforts will help preserve vital carbon sinks and reduce carbon releases. Additionally, implementing proper waste management, promoting recycling, and adopting practices to reduce methane emissions from landfills are essential steps to combat climate change. By taking collective action to curb carbon emissions, society can work towards a more sustainable and carbon-neutral future, ensuring the well-being of current and future generations and protecting the delicate balance of the Earth's ecosystems.

## 2.2. IPCC Regulations' Impact on the CCS Development

Carbon dioxide removals (CDRs) are the most commonly used technologies and practices. These are used to remove $CO_2$ from the atmosphere and store it permanently in different forms, such as underground in geological formations or in the ocean. Many scientists accept that using CDRs is the most effective solution for escalating climate change, as they could help reduce the concentration of greenhouse gases in the atmosphere and limit global warming. Ghiat and Al-Ansari (2021) indicated that there are different types of

CDR technologies and practices available on the market, including natural climate solutions (NCS), bioenergy with carbon capture and storage (BECCS), and direct air capture and storage (DACS) [11]. NCS refers to activities that enhance the natural capacity of ecosystems to absorb and store $CO_2$, such as reforestation and soil carbon sequestration [11]. BECCS involves the use of biomass as a source of energy, with the $CO_2$ emissions captured and stored rather than released into the atmosphere. DACS involves the direct capture of $CO_2$ from the air using specialized machines, which can then be stored underground or used in industrial processes.

It is important to note that CDR technologies and practices are still in the early stages of development and deployment, and there are significant technical and logistical challenges that need to be overcome to provide reliable methods of design in the next decades [12]. On the contrary, many researchers indicate that the long-term cost-effectiveness and sustainability of CDRs are still uncertain, and more research is needed to understand their potential impacts and risks in environmental policy [13]. It must be stressed that the CDRs are increasingly being recognized as an important part of the effort to achieve net zero emissions. These technologies can help implement residual emissions from hard-to-abate industrial sectors and take back already-emitted $CO_2$ [14]. To reduce the detrimental effect of climate change, the United Nations Intergovernmental Panel on Climate Change (IPCC) has stated that the deployment of CDRs will be necessary to achieve net zero emissions and that limiting global warming to 1.5 °C will require the removal of around 6 billion tons of $CO_2$ per year by 2050 through CDR technologies [15]. This enactment has been accepted as a significant policy measure and represents an amount that is equivalent to more than the weight of all petroleum produced today.

Hussin and Aroua (2020) indicated that CDR technologies and practices have a great potential to create co-benefits, including job growth, biodiversity benefits, and equitable growth for the economy, but there are also significant technical and logistical challenges that should be addressed [16]. As previously underlined, CDR technologies have been used to tackle climate change and its impact on the built environment [16]. To this extent, conducting a techno-economic analysis of CDR technologies is essential. Studies indicate that currently most of the CDRs are at an early development stage, and very little research has been undertaken to explore their strengths and weaknesses in the sphere of economics.

Many pilot studies have underlined that the cost of CDR technologies has risen rapidly [17,18]. This rise ranges from USD 250 to USD 600 per ton of $CO_2$, depending on the technology, energy source, and scale of deployment [19]. To address these challenges, there is a need for increased investment in research and development, as well as the deployment of large-scale CDR projects. This could drive down costs and increase the supply of CDRs in order to provide competitive market conditions. To alleviate high investment costs, both the private and public sectors must work together to overcome the challenges and accelerate the development and deployment of CDR technologies to address the global climate crisis [20].

*2.3. Carbon Capture and Storage*

Many countries are taking effective policy measures to mitigate climate change such as introducing mandatory regulations to reduce the use of fossil fuels or recommending the implementation of renewable energy sources [21]. Studies question whether there is any effective control mechanism put into effect for reducing greenhouse gas (GHG) emissions across the globe [22]. For example, one of the pilot studies conducted by Nocito and Dibenedetto (2020) indicated that the installation of wind turbines both on-site and offshore could provide clean energy resources [23]. However, the carbon footprint of wind turbines is questionable because of the high manufacturing process of constructing the turbines in remote locations. Therefore, state-of-the-art technologies are tested to increase CCS systems' capacity and technical specifications to capture and store $CO_2$ to tackle the climate change impact [24].

In fact, CCS (carbon capture and storage) is considered a crucial technology for mitigating climate change by collecting greenhouse gases from the atmosphere and storing them [25]. The International Energy Agency (IEA) report highlights that carbon capture and storage (CCS) could potentially achieve a 19% reduction in carbon dioxide emissions by the year 2050. The IEA also noted that excluding this technology from the conversation on climate change could have the severe consequence of increasing carbon dioxide emissions by 70% [26].

While international and intergovernmental agencies realize the potential of CCS technologies, a couple of concerns need to be addressed if CCS technologies are to be commercialized worldwide [27]. The first concern is the low cost of CCS technology. An affordable and sustainable CCS material should be developed. The second concern is putting mandatory regulations and policies for efficient CCS technology implementation on a large scale [27]. If the technology is feasible and efficient enough, then making it a requirement in industrial applications could have a significant impact on GHG emissions worldwide [10]. CCS technologies have been applied in a variety of industries. According to the Global Carbon Capture and Storage Institute, there are 37 large-scale CCS projects currently in operation. The first CCS project began operating in 1972 in Val Verde, Texas [10]. There are three main types of carbon capture processes, namely, pre-combustion, post-combustion, and oxy-fuel combustion (which includes chemical looping). A flowchart summarizing the carbon dioxide capture and storage process is shown in Figure 2.

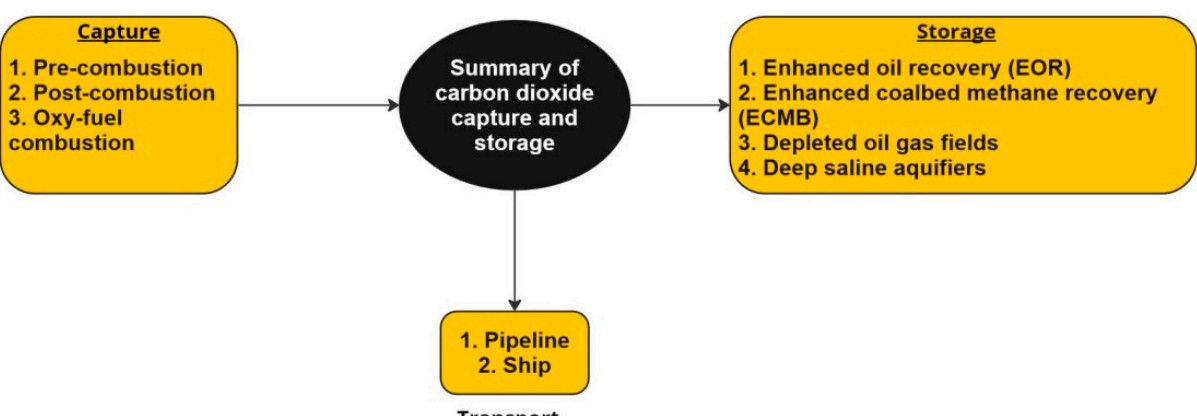

**Figure 2.** Summary of carbon dioxide capture and storage. Drawn by author.

Pre-combustion $CO_2$ capture involves extracting $CO_2$ from fossil fuels prior to completing the combustion process [28]. An example of this technique is found in the gasification process, wherein a feedstock is introduced and partially combusted with steam and air at high temperature and pressure, resulting in the formation of synthesis gas [29]. This syngas is composed of hydrogen, carbon monoxide, and smaller concentrations of other gases. The syngas is then passed through a water–gas shift reaction. The $CO_2$ concentration is around 15–50% [29]. The $CO_2$ can then be captured, stored, and transported. A flowchart that explains the three types of carbon dioxide technologies is shown in Figure 3.

On the other hand, post-combustion $CO_2$ capture deals with removing $CO_2$ from the effluent flue stream [30]. Post-combustion is the simplest technology to use and capture $CO_2$ among the currently available technologies of CSS [30]. There are a couple of ways to capture $CO_2$ from the post-combustion process. A summary of these technologies is presented in Table 2.

Furthermore, various research has been conducted on these technologies to overcome their current limitations and improve them for more wide-scale usage in the future. A summary of the technologies' current challenges and future development is delineated in Table 3.

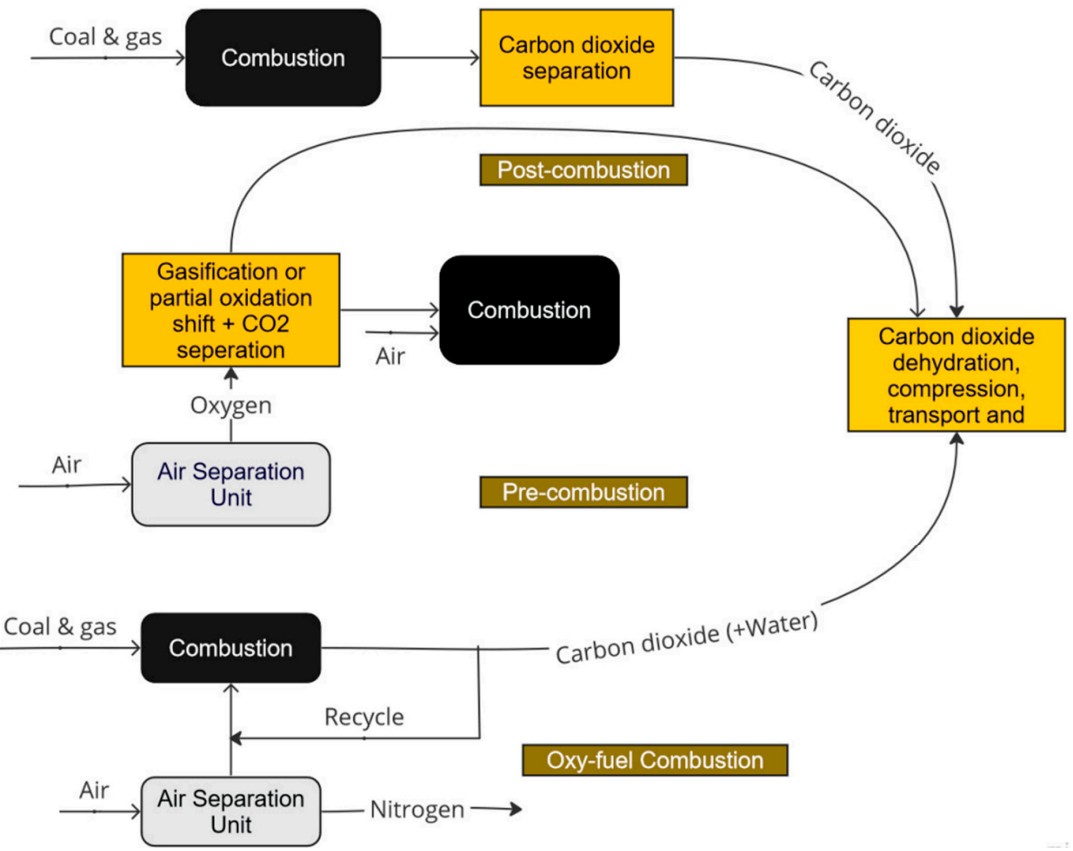

**Figure 3.** Flowchart with the three different carbon dioxide capture technologies illustrated. Drawn by the author.

**Table 2.** A description of various techniques for capturing $CO_2$ after it is produced through the process of combustion.

| Technology Number | CO$_2$ Capture Technologies | Definition |
|:---:|:---:|:---|
| 1 | Absorption | - The process employs an amine-based solvent known as monoethanolamide (MEA), which boasts exceptional reactivity, affordability, and high absorption capacity.<br>- This technology stands as the first of its kind in post-combustion applications that can be globally adopted. |
| 2 | Adsorption | - The adsorption process is advantageous for its reversibility, high adsorption capacity, low energy requirements, and cost-effectiveness of the adsorbent material.<br>- These methods utilize pressure, temperature, electricity, vacuum, or a combination of pressure and vacuum to achieve the desired adsorption effects. |
| 3 | Membrane | - The preference for this method stems from its high selectivity, high driving force, straightforward installation, low capital cost, and minimal energy consumption, even when dealing with low partial pressure of $CO_2$.<br>- During operation, there is often a trade-off between permeability and selectivity that needs to be considered.<br>- The efficiency depends on the driving force created by the $CO_2$ concentration and pressure. |
| 4 | Cryogenic | - Used commercially for exhaust gases that have high pressure and concentration.<br>- Can only be used if high $CO_2$ is present in the stream because of its economic feasibility. |

**Table 3.** A summary of the current difficulties and obstacles that are being encountered in the development and implementation of various techniques for capturing $CO_2$ after it is produced through the process of combustion.

| Technology Number | Technology Name | Current Challenges | Future Suggestions |
|:---:|:---:|:---|:---|
| 1 | Absorption | - Larger absorber volume<br>- Significant energy penalty<br>- High corrosion rate<br>- High energy consumption<br>- Solvent emissions<br>- Additional compression work needed | - Future suggested improvements include:<br>- Developing solvents with high $CO_2$ absorption capacity and non-toxic solvents |
| 2 | Adsorption | - Challenges in selecting good $CO_2$ adsorption material<br>- Low $CO_2$ selectivity<br>- Needs periodic operation<br>- Large pressure drops in flue gas | - Future improvements to commercialize adsorbent technologies are:<br>- High capture performance<br>- High selectivity<br>- Surface modifications are needed<br>- Material should be economical<br>- High $CO_2$ capacity<br>- Simple operation<br>- Environmentally friendly |
| 3 | Membrane | - Requires constant compression<br>- High membrane manufacturing cost<br>- Requires high selectivity<br>- Fouling effect is significant | - Developing new membrane materials<br>- Developing effective membranes for $CO_2$ capture<br>- Further research is needed on transferring labscale research to industrial applications |
| 4 | Cryogenic | - Requires a high amount of energy<br>- Solid $CO_2$ can accumulate on the surface of the heat exchanger, reducing efficiency and affecting heat transfer | - More efficient cryogenic processes need to be developed |

*2.4. Oxy-Fuel Combustion*

Captured $CO_2$ can be purified, compressed, and stored in geological formations. It can also be used in geothermal technology; agriculture; biofuel production, starting with raw materials [31]. Additionally, a closer examination of the various CCS processes has shown that post-combustion is the most applicable technology [32]. It is widely used because of its minimal impact on existing systems. However, it consumes a large amount of energy in the process of separating $CO_2$ gas from the exhaust gases in the air separation unit (ASU). In fact, this process reduces the efficiency of power plants by 8–12%, making it not economically feasible [33].

Another approach that maintains a better efficiency than the ASU is by pre-combusting the gas in an IGCC-constructed plant [33]. However, this method is still unreliable and expensive and requires more development. This is why scientists have concluded that, although oxy-fuel combustion (OFC) requires more research, it is the most promising technology because of its easy $CO_2$ capture and low-efficiency penalty [1]. Furthermore, Porter et al. (2017) stated that large-scale commercialization of oxy-fuel combustion is the best option for achieving near-zero $CO_2$ power generation [34]. Table 4 summarizes the advantages and disadvantages of oxy-fuel combustion technology.

**Table 4.** Summary of advantages and disadvantages of oxy-fuel combustion carbon removal technologies.

| Advantages | Disadvantages |
| --- | --- |
| Has the highest efficiency of carbon capture among the other technologies | High capital investment |
| Reduction in volume of the flue gas | The air separation unit (ASU) requires a high amount of energy |
| Increase in boiler efficiency | Further research is being conducted on oxygen transport membranes, ion-transport membranes, or chemical looping |
| NOx gases elimination | |
| NOx gases elimination | |
| Potential to be used in oxy-fired and IGCC power plant | |

### 2.5. Reviewing Carbon Capture Technologies

Carbon capture and sequestration technologies have been gaining quite a lot of attention recently. As the year 2023 approaches, efforts to meet countries' emission goals are on the rise. CCS technologies provide a golden ray of sunlight during the massive expectations of lowering global $CO_2$ emissions, especially when the question is still being asked if the efforts being carried out by countries are working effectively [35]. CCS is a technology aimed at capturing $CO_2$ emissions from industries and exhaust gases before they can be released into the environment [36]. As industries implement carbon capture technologies, $CO_2$ concentrations in the atmosphere will become less significant. This technology could be applied to large-scale industries, e.g., chemical factories and petrochemical plant exhaust gases, which could capture the $CO_2$ emissions before they could harm the atmosphere [37]. It could also be applied to small-scale carbon emitters, such as car exhaust channels that use petrochemical products as fuel and other emitters, such as air conditioners. Applying carbon capture technologies on a large scale would aid in lowering the amount of carbon being emitted into the atmosphere [38]. An analysis was conducted using VOS viewer software v1.6.18 and the Web of Science website on the keywords of carbon capture technologies, revealing the key areas being investigated by researchers, as shown in Figure 4.

As shown in Figure 4, CCS has been discussed in the literature frequently, starting in the year 2020. Particularly, the topics discussed were performance, storage, adsorption, and post-combustion. Different topics were discussed here, with a special focus on the implementation of adsorption technologies for large-scale utilization.

### 2.6. Waste-to-Energy Technology

The work published by Baena-Moreno et al. (2019) provides an overview of the currently available methods for capturing $CO_2$, with a particular focus on oxy-fuel combustion, which is considered the most reliable technology for carbon capture. Carbon capture is seen as the only technology that can significantly reduce $CO_2$ concentrations in the atmosphere from large-scale energy generation systems [38,39]. Carbon capture consists of pre-combustion, post-combustion, and oxy-fuel combustion. Post-combustion is reported to be the most mature technology for carbon capture, but its main disadvantage is the significant energy loss that results from separating $CO_2$ from exhaust gases [40].

The scientific community agrees that developing oxy-fuel combustion technology and net-zero carbon in power generation sectors is needed [41]. In addition, the advantages of using oxy-fuel combustion include a reduction in the volume of flue gas, an increase in boiler efficiency, the elimination of NOx gases, and the stabilization of temperature. However, scientists have reported that the problems associated with this technology include high capital investment and energy consumption of the air separation unit (ASU) [42]. As

a result, further investigations are required into new air separation methods, such as oxygen transport membranes, ion-transport membranes, or chemical looping. Among these, chemical looping has the potential to significantly improve oxyfuel [43]. The diagram shows a flow chart of an oxy-fuel combustion process in Figure 5.

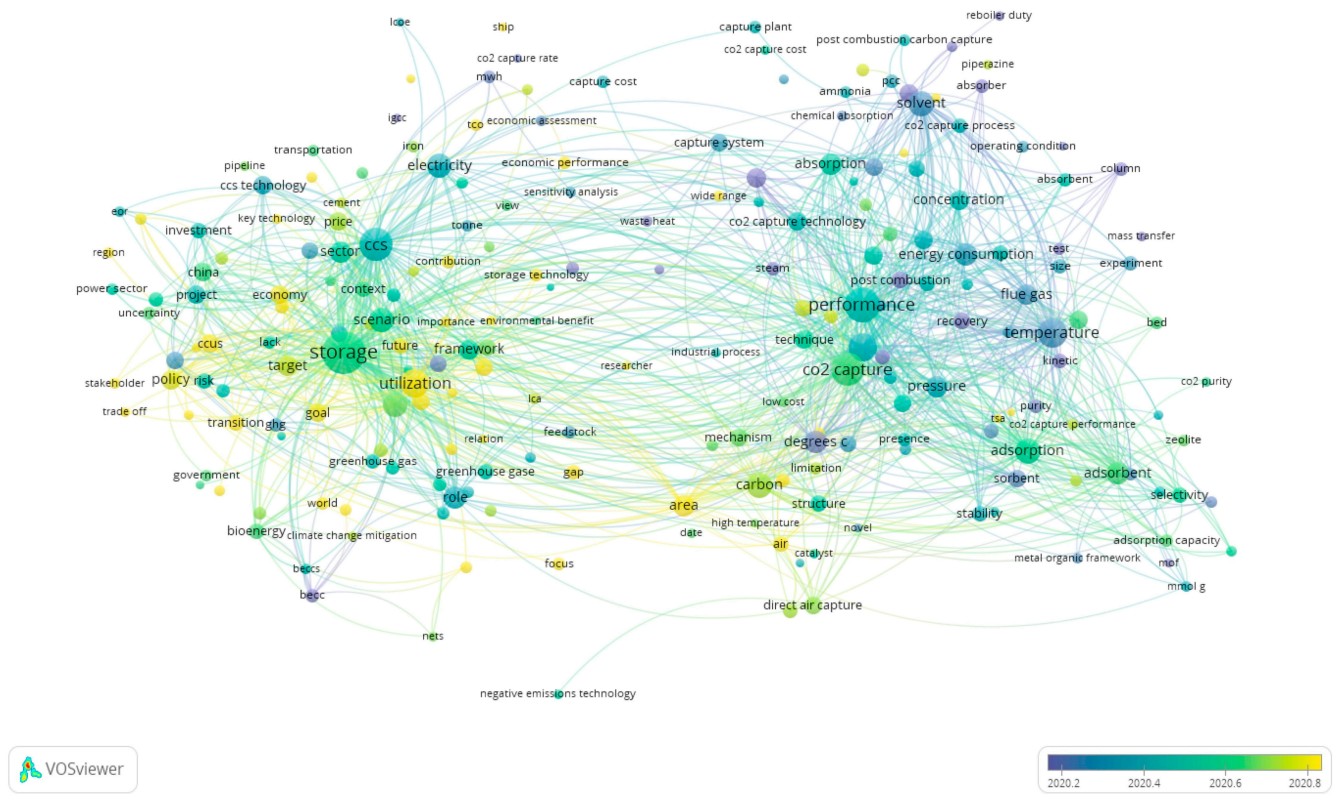

**Figure 4.** Carbon capture and sequestration analysis from VOS viewer. Data source: https://www. vosviewer.com (accessed on 25 January 2023).

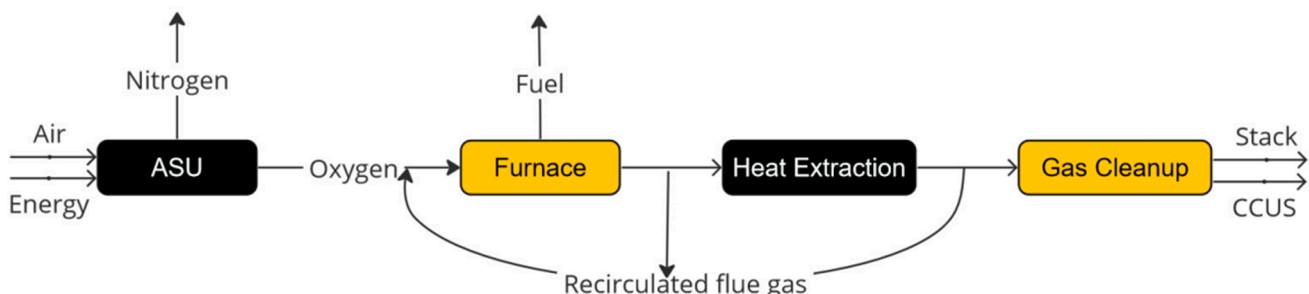

**Figure 5.** Flow diagram of the oxy-fuel combustion process. Drawn by author.

Oxy-fuel combustion systems have recently gained momentum as a cost-effective and technically feasible method for carbon capture [44]. Oxy-fuel combustion replaces air with high-purity oxygen as the oxidant in concentrated flue gas, and coal has so far been the primary fuel type studied [44]. However, biomass has recently been considered a potential energy source due to the depletion of non-renewable resources. Biomass has significant potential, particularly for countries with limited natural resources. Using biomass as an energy source could reduce their reliance on energy imports and increase their energy security [45].

The pilot study conducted by Porter et al. (2017) illustrates that oxy-fuel combustion can be implemented in existing power plants as well as in constructing new plants, and it can use different fuels such as biomass or municipal waste [34]. In addition to producing

energy, this process has the potential to eliminate waste landfills and non-recyclable materials. The only concern raised by scientists is the high energy cost of the ASU unit during the purification of the air stream. It is estimated that global waste material will increase from 2.01 billion tons to 3.4 billion tons. Figure 6 shows the global waste composition.

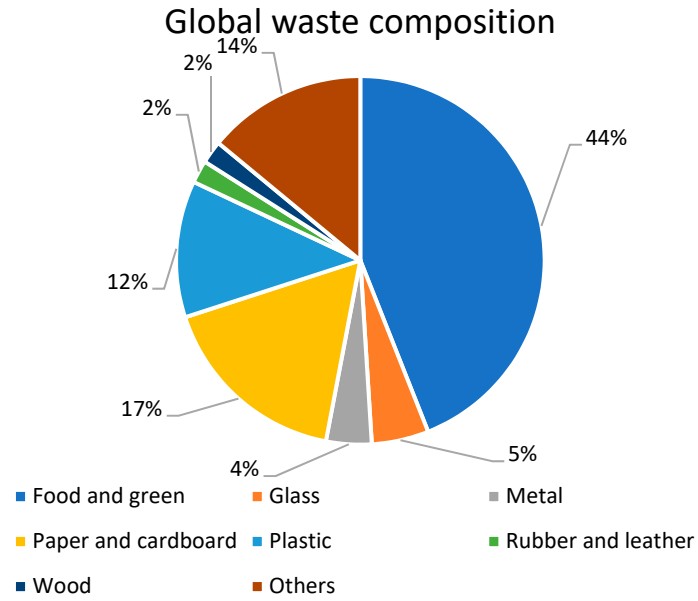

**Figure 6.** Global waste composition as potential energy resource. Drawn by author.

Biomass Utilization in W2E Applications

Biomass serves as a carbon sink, absorbing $CO_2$ during growth. When used as an energy source, it can be carbon-neutral. Coupled with carbon capture and storage (CCS) technology, biomass power generation becomes carbon-negative, removing more $CO_2$ than it emits. This approach offers a sustainable and effective way to combat climate change by reducing atmospheric carbon levels. Sustainable practices and continued CCS advancements are crucial for maximizing its potential [46].

In addition, a new technology that is rapidly gaining momentum and that uses biomass as an energy source and incorporates carbon capture and storage is bioenergy with carbon capture and storage (BECCS or BioCCS). This process combines two $CO_2$ mitigation processes to achieve net-zero or negative $CO_2$ emissions [47]. In addition to illustrating the cruciality of BioCCS, the pilot study has found that the total $CO_2$ emissions for oxy-fuel combustion of biomass are $-0.2722$ kg $CO_2$/MJ, whereas emissions from municipal solid waste incineration integrated with CCS systems are around $-0.7$ kg $CO_2$/kg of wet MSW. This demonstrates that the BioCCS process can significantly contribute to the decarbonization of Europe. According to Giannousakis et al. (2021), there are approximately 20 BECCS projects worldwide that are related to bioenergy technologies [9]. Table 5 demonstrates the list of countries and examples of BECCS projects.

As can be seen in Table 5, this system is efficient and technologically feasible. However, the main drawbacks to commercializing this technology are the lack of effective policies and economic regulations for using bioenergy with carbon capture. It should be noted that there are many modern incineration plants in operation today that must not only comply with environmental regulations but also be accepted by the public authorities [48]. As a result, more plants are being designed to be environmentally friendly. An example of such a plant is the combined heat and power (CHP) waste incineration plant in Copenhagen. This plant has high-efficiency performance and also includes a recreational area with a ski resort, viewpoint, and climbing wall. While incineration is looked down on as a process for waste removal, it could be an alternative option if it were coupled with carbon capture technologies. There are currently four incineration plants that are coupled with carbon

capture technologies in the world, which are located in Norway and Japan, and there are two plants in the Netherlands. While incineration is often viewed negatively as a waste management method, it can be an economically viable option when coupled with carbon capture technologies [49]. These plants are using various carbon capture technologies, such as an alkaline aqueous amine method introduced by Toshiba in Japan, which is used to collect $CO_2$ for local cultivation and algae culture formation.

**Table 5.** Countries that implement BECCS into their plants.

| Country | Technology |
|---|---|
| Norway<br>The Netherlands | Waste-to-energy (WtE) |
| France<br>Brazil<br>Sweden | Ethanol plants |
| Japan<br>Sweden<br>The United States<br>Sweden | Biomass combustion and co-firing<br>Pulp and paper plants<br>Biomass gasification<br>Biogas plant |

Reprinted with permission from [9]. Copyright 2021 Elsevier.

### 2.7. Carbon Dioxide Separation Technologies

Mukherjee et al. (2019) explained that there have been multiple summaries of patent applications on $CO_2$ captures published, with a focus on various technologies such as absorption, adsorption, membrane, cryogenic, enzymatic, and hybrid technologies [50]. Additionally, more research has been conducted on the different types of solvents, sorbents, and membranes that can be used in these technologies, as shown in Figure 7 [51]. However, due to the lack of adsorption processes in CS, this study aimed to review recent publications made between 2014 and 2018 to fill the knowledge gap between laboratory research and industrial applications. The study also aims to provide insights into recently published articles about innovative adsorption technologies that can be used economically.

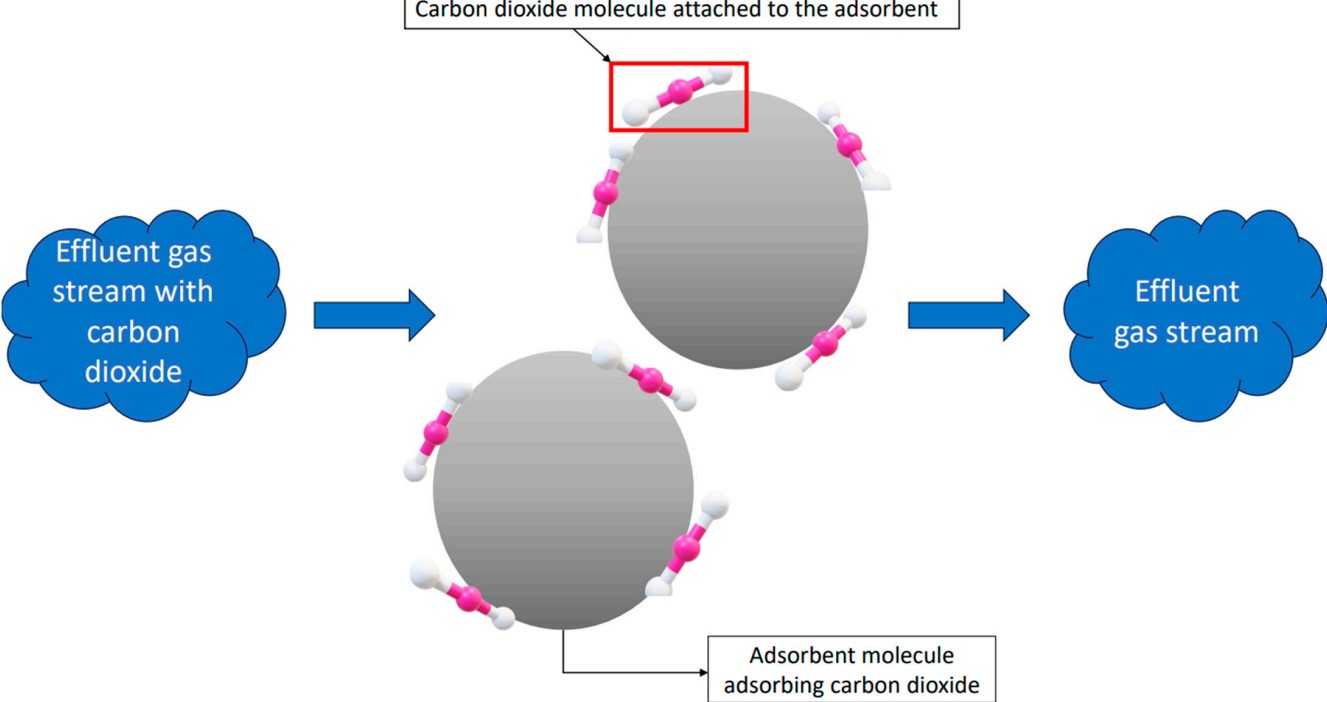

**Figure 7.** Adsorption process applied to carbon capture technologies. Drawn by author.

The main challenge faced by developing countries in implementing carbon capture technologies is the development of suitable adsorbents that have certain properties to be used for CS. Some of these properties include capacity, economy, recyclability, and low production costs [52]. The adsorbent must also have sufficient sustainability in terms of multiple sorption and desorption cycles, a low cost of production, and widely available raw materials. The technical feasibility of the used adsorbent must also be thoroughly assessed, considering factors such as its mass transfer ability, fast kinetics of $CO_2$ sorption and desorption, and the type of bed used. The cited work concluded that significant improvement is needed in the field of carbon capture to achieve substantial $CO_2$ capture [53].

In summary, carbon capture technologies using adsorption processes have gained significant attention in recent years as an alternative solution to $CO_2$ emissions. These technologies involve the use of adsorbents, which are materials that can adsorb (or bind) gases on their surface to capture $CO_2$ from various sources, such as power plants and industrial processes. While there have been many patents filed and research published on these technologies, there is still a need for further development in order to make them more practical and cost-effective. In particular, there is a need to improve the sustainability and technical feasibility of adsorbents, as well as to consider other factors such as mass transfer and desorption kinetics when selecting the most appropriate technology. To sum up, this study highlights the importance of investing in and developing carbon capture technologies in order to mitigate the negative impacts of $CO_2$ emissions in the built environment.

*2.8. Carbon Sequestration and Utilization*

Capturing $CO_2$ entails many various steps in order to strip the gas from the effluent stream of many industrial exhaust gases to avoid the gas being released into the atmosphere. However, there follows the problem of what to do with the captured gas. Researchers [54–58] have researched the possible and potential ways to store and utilize that captured carbon. Suggested ways include storing the gas underground and utilizing $CO_2$ in other industries.

Geological storage occurs after capturing carbon dioxide from the exhaust gases; it is then transported to different locations either to keep the gas from leaking or to use it for some other application [59]. This section will discuss the different ways in which captured $CO_2$ is dealt with. Captured carbon dioxide ($CO_2$) can be stored in several ways, with the primary methods being geological storage and industrial utilization. Here are the main storage options:

Geological Sequestration and Storage:

$CO_2$ storage in abandoned geological formations provides a promising solution for mitigating greenhouse gas emissions. The selection of a suitable site requires careful consideration of specific criteria, as described by Bachu [60]. Key factors include the site's capacity, porosity, thickness, accessibility, sealing capability, and geological stability. While geological storage methods can be adapted from existing processes, such as enhanced oil recovery (EOR) projects, there is currently limited practical experience on a commercial scale [61]. Additionally, understanding the potential long-term environmental impacts of storing large quantities of $CO_2$ remains limited. Different geological storage sites are considered, including abandoned oil and gas reservoirs, unmendable coal beds, and saline aquifers. While deep ocean storage is an option, environmental concerns surrounding ocean acidification and eutrophication make it less favorable.

- Criteria for Selecting $CO_2$ Storage Sites:

  $CO_2$ storage sites are chosen after careful evaluation of various factors:

  - Capacity of the Site: Adequate capacity is crucial to accommodating significant $CO_2$ emissions.
  - Porosity: High porosity facilitates effective $CO_2$ storage within the geological formations.
  - Thickness: A thick geological formation provides a larger volume for $CO_2$ storage.
  - Accessibility: Easy access to the site is essential for practical implementation.

- Sealing Capability: The site's ability to prevent $CO_2$ leakage is critical to avoiding environmental risks.
- Geological Stability: A stable geological formation ensures long-term containment of $CO_2$, ensuring safety.

- Adaptation from Enhanced Oil Recovery (EOR) Projects:

Geological $CO_2$ storage methods draw insights from EOR projects, which involve injecting fluids into the subsurface. However, implementing this approach on a commercial scale is currently limited due to the need for further experience and understanding of unique challenges.

- Limited Practical Experience and Environmental Impacts:

The lack of substantial practical experience with large-scale $CO_2$ storage presents challenges to its widespread adoption. Ongoing research and pilot projects are necessary to assess feasibility and effectiveness.

- Different Geological Storage Sites:
  - Abandoned Oil and Gas Reservoirs: Empty or near-empty reservoirs are considered potential $CO_2$ storage sites.
  - Unmendable Coal Beds: Certain coal beds that are unsuitable for mining may serve as storage sites.
  - Saline Aquifers: Underground saline water formations offer another option for $CO_2$ storage.
  - Deep Ocean Storage: While a potential storage solution, it raises environmental concerns related to ocean acidification and eutrophication, making it less encouraged [62].

$CO_2$ storage in abandoned geological formations holds promise for climate change mitigation. The criteria for site selection play a crucial role in ensuring successful storage. Despite drawing from EOR projects, practical experience at a commercial scale remains limited, and long-term environmental impacts require further investigation. Considering various geological storage options while addressing environmental concerns is essential for sustainable $CO_2$ storage initiatives.

Industrial Utilization:

- Enhanced Oil Recovery (EOR): $CO_2$ can be used to enhance oil recovery from oil fields. The injected $CO_2$ helps to mobilize oil, making it easier to extract. While this method captures and stores some $CO_2$, it also increases oil production and, consequently, carbon emissions from burning the additional oil [63].
- Mineralization: $CO_2$ can be converted into mineral forms by reacting with certain rocks or minerals. This process, called mineral carbonation, results in the stable, long-term storage of $CO_2$ in solid carbonate minerals [64].

Furthermore, it is essential to ensure that the stored $CO_2$ remains secure and does not leak back into the atmosphere, as this would negate the purpose of carbon capture and storage efforts. Proper monitoring and regulation are critical to maintaining the integrity of the storage sites. Additionally, continued research and innovation are essential to improving the effectiveness and safety of carbon capture and storage technologies.

Biological Sequestration and Storage:

One promising approach that harnesses the power of nature is biological carbon sequestration. This method involves leveraging living organisms, such as plants, trees, and marine ecosystems, to capture and store carbon dioxide ($CO_2$) from the atmosphere. Through photosynthesis and other biological processes, these natural systems absorb $CO_2$, transforming it into organic matter and locking it away for extended periods. In this section, we will explore the various biological carbon sequestration methods and their potential to play a significant role in combating climate change [65].

- Afforestation and Reforestation

Afforestation and reforestation are powerful techniques to enhance carbon sequestration. Planting new forests or restoring depleted ones allows trees to act as carbon sinks by absorbing $CO_2$ during photosynthesis. The captured carbon is then stored in the trees' roots, stems, and leaves. Such efforts not only contribute to atmospheric carbon reduction but also foster biodiversity and ecosystem restoration [65].

- Forest Management

Beyond creating new forests, effective forest management practices are crucial for maximizing carbon sequestration. By curbing deforestation, implementing sustainable logging, and protecting old-growth forests, we can enhance their capacity to capture and store carbon.

- Agroforestry

Agroforestry blends agriculture with tree planting, providing a win–win scenario for carbon sequestration and sustainable food production. Combining crops and trees on the same land allows for increased carbon uptake while also improving soil health and resilience to climate change.

- Soil Carbon Sequestration

Soil plays a pivotal role in carbon storage. Adopting soil carbon sequestration practices, such as cover cropping, reduced tillage, and agroecological farming, can boost organic carbon levels in the soil, effectively sequestering $CO_2$.

- Wetland Restoration

Wetlands, such as marshes and swamps, are natural carbon sinks. Restoring and preserving these ecosystems helps maintain their carbon sequestration potential while promoting biodiversity and mitigating the impacts of rising sea levels.

- Blue Carbon Ecosystems

Blue carbon ecosystems, including mangroves, seagrasses, and salt marshes, offer substantial carbon sequestration benefits. These coastal habitats sequester large amounts of carbon in their sediments, making them valuable allies in the fight against climate change.

- Biochar

Biochar, a charcoal-like substance produced through pyrolysis of organic biomass, can lock carbon in a stable form. Adding biochar to the soil enhances its carbon sequestration capacity, offering a long-term storage solution for captured $CO_2$.

- Carbon Farming

By implementing carbon farming practices, such as crop rotation and managed grazing, agricultural lands can become efficient carbon sinks. Integrating these strategies with traditional farming methods contributes to increased carbon sequestration potential.

- Marine Algae and Phytoplankton

The vast oceans hold significant potential for carbon sequestration. Certain types of marine algae and phytoplankton absorb $CO_2$ during photosynthesis, playing a critical role in capturing atmospheric carbon and supporting marine ecosystems.

Biological carbon sequestration methods offer a ray of hope in the fight against climate change. By harnessing the natural abilities of plants, trees, and marine ecosystems to capture and store carbon, we can significantly reduce greenhouse gas emissions and mitigate the impacts of global warming. These techniques not only contribute to a healthier environment but also promote ecological restoration, biodiversity, and sustainable land and marine management practices. However, it is crucial to remember that while biological carbon sequestration is a valuable tool, it must complement broader efforts to reduce human-caused emissions. Embracing these natural solutions and combining them with

emission reduction strategies is the key to securing a sustainable and resilient future for our planet.

### 3. Methodology

#### 3.1. Conceptual Framework

The study adopts a mixed-method design approach to provide a global overview of carbon capture technologies from technical and efficiency perspectives. The most efficient carbon removal technologies were investigated in terms of carbon absorbance, mass separation from the solvent, etc. The quantitative analysis considered economic evaluations of the study, considering energy and climate modeling.

In this study, a comprehensive analysis was conducted to identify the most efficient carbon capture technology currently available on the market, as shown in Figure 8. The analysis would evaluate parameters such as carbon absorbance, mass transfer rate, and efficiency of separation, including $CO_2$ inlet and outlet concentrations. This analysis was conducted by following up on recent literature available on carbon removal technologies. First, an equal baseline model was developed. This was conducted by reviewing the differences between the inlet stream to the carbon removal process unit and an outlet stream coming out of the process unit, as shown in Figure 9.

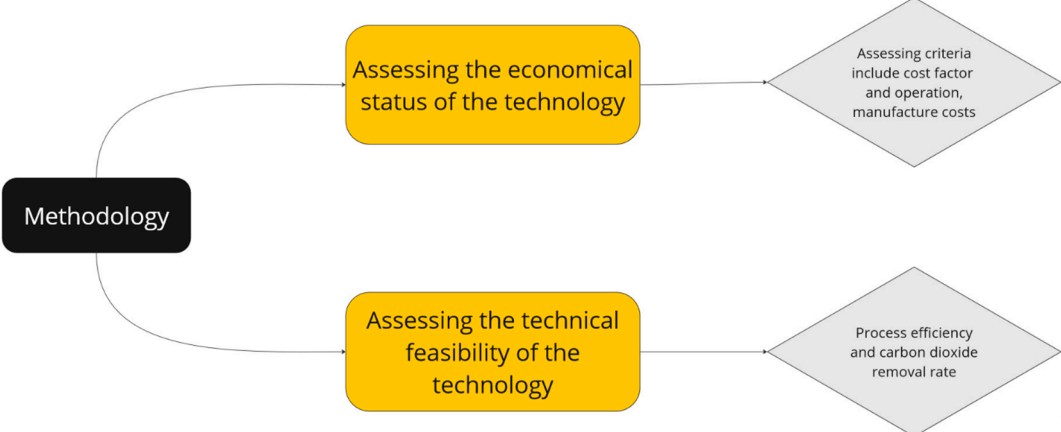

**Figure 8.** Methodology framework flow chart. Drawn by author.

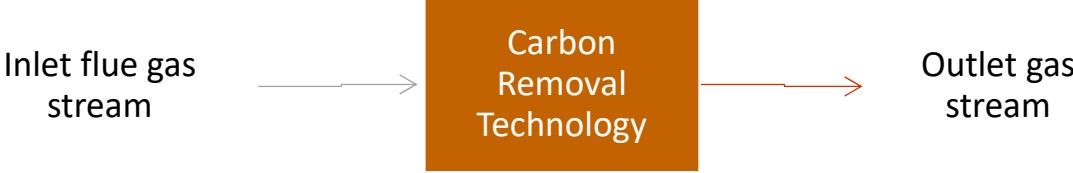

**Figure 9.** Carbon removal separation calculation flow chart. Drawn by author.

The $CO_2$ concentration was calculated before and after the process unit. Then, the concentration was compared, and a separation efficiency percentage was calculated. This efficiency calculation demonstrated the efficiency factor. It must be stressed that the data analysis is dependent on collecting information about the technologies from the literature and performing an analysis using flue stream data from an industrial plant.

The quantitative analysis consisted of multiple parts. An analysis was conducted to measure the performance of the $CO_2$ removal processes. Additionally, the economic appraisal of carbon capture technologies was investigated. Calculations consisted of conventional economic analysis methods, namely, Levelized Cost of Energy (LCOE), Net Present Value (NPV), Present value (PV), etc. Furthermore, these calculations would be useful in finding the carbon removal technology with the highest potential [66].

The economic analysis of the carbon removal technology is subject to a wide variety of inconsistencies and uncertainties, such that it is not sure that a valid method or a proper conclusion can be drawn from the analysis. Moreover, this analysis is subject to inconsistencies from one country to another based on to what extent a certain technology is commercialized. In fact, this is a common inconvenience in achieving renewability worldwide because technologies are not quite reliable yet [67]. Furthermore, the cost of carbon worldwide and carbon taxes have shown differences from one developing country to another depending on the amount of $CO_2$ emissions worldwide [1]. According to the Paris Agreement, this tax is changing, and countries must pay if they are not within the minimum $CO_2$ emission range. In fact, these reasons are one of the biggest barriers to the commercialization of this technology and other renewable technologies worldwide. In Table 6, both the technical and economic parameters are delineated. These indicators consisted of using methods from both qualitative and quantitative analyses.

**Table 6.** Carbon capture indicators for reliable assessment.

| Analysis Type | Criteria to Be Found |
| --- | --- |
| Technical aspect analysis | Inlet carbon dioxide concentration |
| | Outlet carbon dioxide concentration |
| | Separation efficiency calculation |
| | Mass transfer rate |
| | Climatic condition |
| | Minimum carbon dioxide concentration required for each technology |
| | Life cycle assessment |
| Economic analysis aspect | LCOE |
| | NPV |
| | PV |
| | Whether the project is worth it economically |

In risk assessment issues, the decision-making process is based on the utilization of different quantitative models [68]. This is conducted by assessing the different individual parameters that make up the essential information to determine whether the risk is individual or constitutes many different factors [68]. When it comes to choosing an accurate carbon capture and sequestration technology, different analyses are required for reliable decision making. First, an analysis based on the technology's technical feasibility assessment is required [69]. Moreover, analysis of the $CO_2$ concentration in the flue gas is required to determine the type of technology that needs to be used. Another parameter that should be considered is the economic cost of the technology [70]. Economic costs, including affordability and availability, are required for the identification of appropriate technologies [71]. This is because even if a suitable technology is developed, it may be too costly or too difficult to manufacture effectively while considering current carbon capture technologies. To this extent, this study aimed to develop a reliable conceptual framework for the implementation of carbon capture technologies.

*3.2. Effect of Technology Costs on Mitigation Factors*

The methodology was adopted in determining the decision-making process to support the utilization of quantitative models [72]. By collecting data, scientists are able to analyze whether certain parameters are relevant to carbon capture technologies. From the work of Borgonovo et al. (2010), the Probabilistic Safety Assessment (PSA) was developed. It analyzes individual changes based on the interaction of different parameters [73]. Gardarsdottir et al. (2019) discussed the uncertainty in technology prices related to $CO_2$ removal availability and climate change mitigation pathways and how it affects decision-making. In fact, Hepburn et al. (2019) implemented the PSA analysis for their methodology [74–78].

To implement carbon dioxide removal (CDR) technologies in industries these days, an effective analysis of their technical and economic feasibility is required. Giannousakis (2021)

conducted a study on the effect of the uncertainty of CDR technologies and performed a sensitivity analysis on these scenarios with and without implementing them and considered how their cost would be affected [9]. The energy-economy climate model REMIND was used to assess the effect of energy prices and carbon removal availability, which has affected mitigation factors [9,76]. Fikru (2022), on the other hand, discussed that there are different climate policies with different goals that lead to different carbon implementation techniques, and it is crucial to implement them in analyses [76]. The policies range from no-policy to different politically accepted climate policies. To implement the different political scenarios, Giannousakis (2021) proposed two different reliable, economically viable scenarios, as shown in Table 7.

**Table 7.** Different policy scenarios.

| | With CDR | Without CDR | Description |
|---|---|---|---|
| No-policy baseline | $3 + 9 \times 4$ | $3 + 9 \times 4$ | No climate policy |
| NDC | $3 + 9 \times 4$ | $3 + 9 \times 4$ | No increased ambition |
| B1300 | $3 + 9 \times 4$ | $3 + 9 \times 4$ | 67% prob. of 2 °C |
| B1100 | $3 + 9 \times 4$ | $3 + 9 \times 4$ | Well below 2 °C |
| B900 | $3 + 9 \times 4$ | $3 + 9 \times 4$ | 67% prob. of 1.5 °C |

Reprinted with permission from [9] Copyright 2021, Elsevier.

It is important to note that the cost of $CO_2$ removal (CDR) technologies plays a significant role in climate mitigation pathways [77]. In this study, the authors analyzed nine different techno-economic parameters to assess the mitigation potential of CDR technologies, including the costs of renewable energy technologies, nuclear power plants, fossil fuel extraction, biomass supply, and the rate of injecting $CO_2$ into the ground as a CDR technology. Yong et al. (2022) also considered the costs of electric and hydrogen-fueled vehicles, as they can serve as alternatives to oil in the transport sector.

Yong et al. (2022) conducted their analysis using scenarios with and without the implementation of CDR technologies to assess the availability, price, and feasibility of these technologies in different climate policies [78]. These policies included the no-policy baseline, the currently announced Paris pledges, and three additional scenarios. The results of this analysis showed that CDR technologies, such as BECCS, have the potential to significantly contribute to decarbonization, particularly when coupled with renewable energy technologies and other CDR technologies. However, Yong et al. (2022) also noted that there is a need for further development and improvement in CDR technologies in order to make them more economically viable and technically feasible [78].

### 3.3. Energy Generation Methods with $CO_2$ Removal

Oxyfuel combustion is a process that uses oxygen instead of air for combustion to produce a concentrated stream of $CO_2$ in the flue gas [79]. This process has been widely used in the incineration of waste materials, as it allows for higher processing temperatures and can potentially reduce fuel consumption. Additionally, oxyfuel combustion can increase the efficiency of boilers by reducing the amount of air required for combustion, leading to lower investment costs [79]. However, there is a lack of research on the use of oxyfuel combustion in waste incineration. One method used to study the thermal decomposition of different types of waste is thermogravimetric analysis (TGA), which can be combined with different gas-analyzing technologies [80]. Some of these technologies are gas chromatography (GC), mass spectrometry (MS), and sometimes Fourier transform infrared (FTIR) spectroscopy is used to analyze the gaseous products produced during the process.

It is important to note that the main advantage of using oxyfuel combustion in waste incineration is the production of a concentrated stream of $CO_2$, which can then be captured and stored [81]. This can significantly reduce the overall $CO_2$ emissions of the incineration process. However, there are also challenges associated with this technology, including the high energy consumption of the air separation unit (ASU) needed to produce the high-purity oxygen required for the process [82]. Additionally, the high capital investment

and technical complexity of implementing oxyfuel combustion may also be barriers to its widespread adoption [83]. Despite these challenges, more research is required to be conducted on this technology, as it holds promise for achieving near-zero $CO_2$ emissions in power generation and waste management. Certain technologies are being implemented to explore processes such as pyrolysis and thermal decomposition using devices such as a thermogravimetric analyzer, specially designed spectroscopy, etc., to analyze the processes' formation [84].

Petrochemical wastewater is a type of waste material that can be harmful and cause environmental pollution if not properly stored and contained [85]. Pyrolysis is a method that can be used to treat and dispose of petrochemical wastewater. During the pyrolysis process, the wastewater is subjected to thermal decomposition in a carbon dioxide ($CO_2$) and nitrogen ($N_2$) environment [86]. At low temperatures, $CO_2$ acts as an inert gas. Researchers have identified three stages in the pyrolysis of wastewater in a $CO_2$ environment: drying, devolatilization, and char–$CO_2$ gasification at high temperatures. In contrast, when the pyrolysis process is carried out in an $N_2$ environment, only two stages are observed: drying and devolatilization. This is because the absence of $CO_2$ prevents char formation.

The analysis of various pyrolysis methods has shown that pyrolysis can be used to dispose of waste materials while also generating energy [87]. Researchers have also examined the impact of different environments on the pyrolysis process. At lower temperatures, the results of pyrolysis show minimal variation. However, as temperatures rise, carbon dioxide ($CO_2$) starts to play a significant role, acting as a gasification agent that interacts with the char. This char–$CO_2$ gasification process is highly endothermic, necessitating a substantial amount of energy, and is exclusive to elevated temperatures [88].

## 4. Analysis

### 4.1. Overview of CCS Research Publication Trends

Figure 10 shows the categorization of the various CCS types and the recent research that has been conducted in this area. As shown in Figure 10, each of the bigger circles illustrates a trending keyword that is commonly used in the literature. The bigger the circle, the more research has been conducted for the selected keyword.

In addition, Figure 10 illustrates that very little research has been conducted in the field of biomass and solar energy in the background of CCS technologies' implementation. For future work, more research should investigate this area. Terms such as renewable energy, solar energy, biomass, and gasification are all connected terminologies to carbon dioxide emissions in the background of CCS implementation. Coal combustion is the determinant contributor to $CO_2$ levels in the atmosphere, including fossil fuels [89]. These terms are connected to post-combustion. The important finding from these data is that they are all connected to biomass, solar energy, and gasification, although a weak link is shown, which indicates that little research has been conducted in this field.

Furthermore, upon taking into consideration the previous keywords, another relationship can be obtained. This illustrates the relationship between different CCS technologies and some methods that are used in analyzing CCS technologies. Some of these methods include implementing computer simulations, energy security, life-cycle assessment, energy conversion, controlled study, renewable energy, commerce, sustainability, energy storage, decision-making, and comparing studies, which are all keywords that showed up when an analysis was performed on the recent CCS publications and the technologies and methodologies that researchers used in their studies. That is why it is crucial to investigate whenever a CCS topic is discussed.

### 4.2. CCS Knowledge Structure

There is a relationship between $CO_2$ emissions, population, energy efficiency, etc., according to Ryu et al. (2022). The equation for this relationship is:

$$CD = P \times \frac{GDP}{P} \times \frac{E}{GDP} \times \frac{C}{E} - S_{CO_2}$$

where CD is the carbon emissions, GDP is the economic development, P is population, E is the energy production, C is the carbon-based fuel used for energy production, and $S_{CO_2}$ is the $CO_2$ sinks [90]. As can be seen, there are three ways in which carbon levels can be decreased. For example, by increasing the efficiency of energy production (E/GDP), changing fossil fuel-based fuels into more carbon-free fuels, and increasing the size of carbon sinks around the world. The third part of increasing carbon sinks can be achieved by developing CCS technologies, such as pre-combustion carbon removal, post-combustion removal from the flue gas, and oxy-fuel combustion (OFC), and also chemical looping and clathrate hydrate processes [2].

There are different methods by which CCS technologies are assessed, and each type of CCS is assessed differently [75]. Particularly, CCS is more efficient with flue streams that have high $CO_2$ concentrations; otherwise, the CCS technology may not be efficient. In fact, $CO_2$ concentration in flue gas is a major roadblock to applying carbon capture technologies [91].

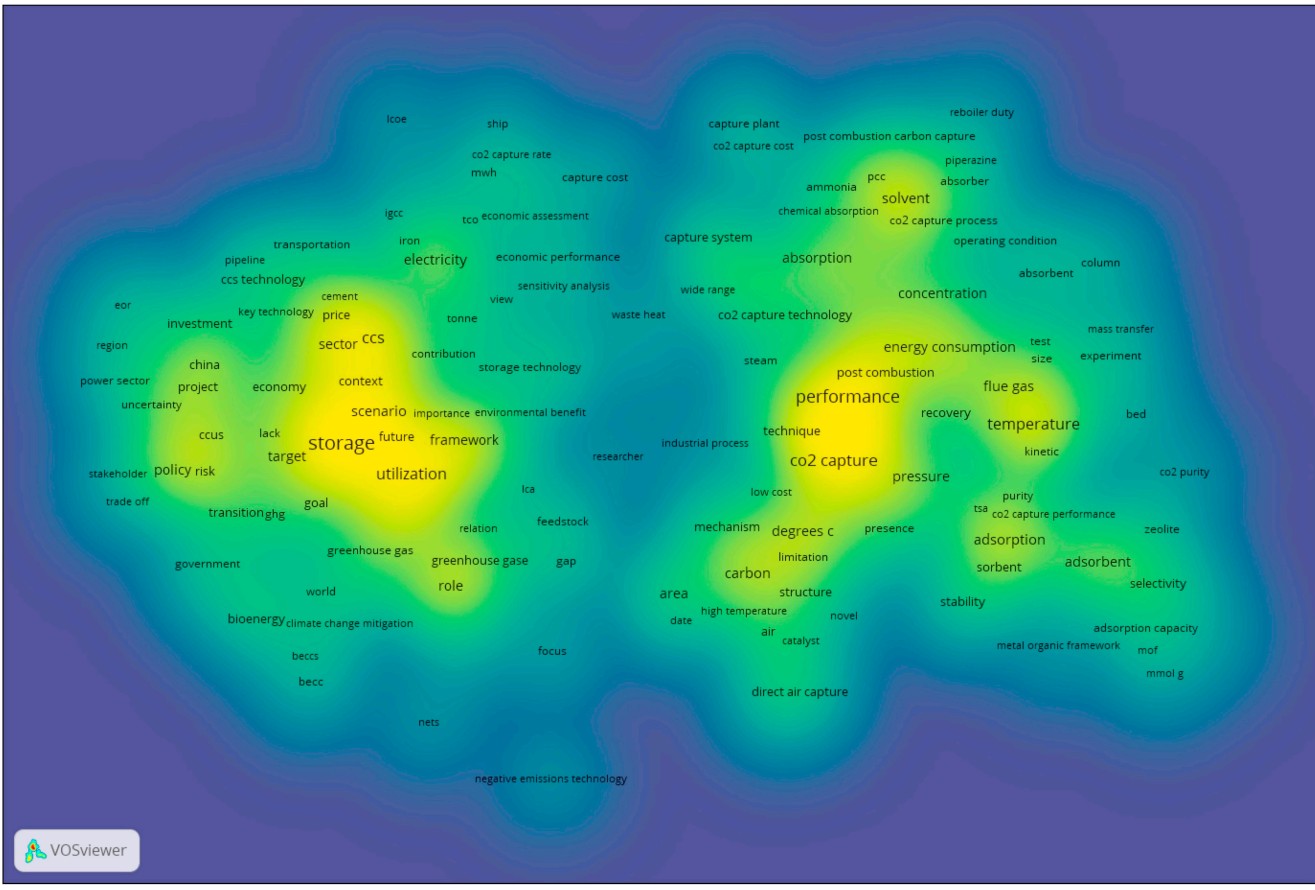

**Figure 10.** VOS viewer illustrates CCS types in their categories. Data source: https://www.vosviewer.com (accessed on 25 January 2023).

### 4.3. Modeling and Kinetic Studies

There are two methods available to assess $CO_2$ absorption through the equilibrium stage and rate modeling concepts [92]. Rate modeling methods are used more than the equilibrium stage method. For example, this method was used by Mukherjee et al. (2019). This pilot study investigated a packed column by using MEA as the absorbent, and then the study conducted the analysis using second-order kinetics [50]. In addition, Agaton (2021) studied the rate fraction in Aspen Plus for systems using ammonia and AMPPZ blended absorbents [27]. Chao et al. (2021) worked on a similar method of design by using Aspen plus simulations to assess (AMP + PZ) absorptivity [28]. Furthermore, Mukherjee et al. (2019) reviewed different models that were used in analyzing the post-combustion type of

$CO_2$ using amines [50]. In addition, models that involved two film theories were found to be more accurate when it came to analyzing absorber efficiency in comparison to the penetration theory [23]. There are multiple software platforms that are successful in performing analysis with modeling. For example, some platforms, including gPROMS, Aspen- Plus, Customer modeler, Plus Dynamics, MATLAB with ODE solver, and Simulink, were chosen to conduct the simulation analysis thoroughly. Additionally, computational fluid dynamics (CFD) models were employed to understand the hydrodynamic phenomenon. Sarkar et al. (2014) used computational fluid dynamics tools to model and analyze the performance of different carbon capture technologies [70].

*4.4. Some Tools That Are Used in CCS Analysis*

While discussing CCS technologies, certain tools and methods should be investigated to provide reliable economic assessments [93]. An example of such tools includes computer simulations that help analyze the different workings of different technologies in different environments. Such simulations help in coming to conclusions related to the working principles of a specific technology. They may also aid when it comes to deciding between two potential technologies that may be suggested. Different examples of mathematical modeling and simulations are shown below.

I.    FOQUS: A structure is proposed for optimizing and measuring uncertainty and sensitivity, facilitating the integration of fundamental data sub-models like thermodynamics and kinetics into comprehensive process models. This integration allows for the swift synthesis and optimization of processes while also assessing the degree of uncertainty linked to the final outcomes.

II.   COPLOS, which stands for 'communication about prospects and limitations of simulation results for policymakers', is an exchange between simulation experts and simulation users who work in the field of carbon capture technologies. Different research articles have been developed using COPLOS as their tool of analysis. This tool is helpful and is used to bridge the gap between technical experts and policymakers. This tool has the potential, if used frequently, to speed up the development of different CCS, and its commercialization potential could be thoroughly developed.

III.  PIMS stands for polymers of intrinsic microporosity (PIMS), which are a unique class of polymers that are commonly used to capture $CO_2$. Such polymers are used to assess and analyze the carbon capture potential of a post-combustion CCS technology, for example, absorption technology. This technology is still in development, but it provides a reliable measure of how efficient and applicable a certain technology is when it comes to absorbing $CO_2$ from the atmosphere.

*4.5. Economic Analysis of CCS Technologies*

As the world grapples with the escalating threats of climate change, the imperative to mitigate carbon dioxide ($CO_2$) emissions has never been more urgent. In response, carbon capture technologies have emerged as vital solutions to address this pressing global challenge. Among these innovations, post-capture, pre-capture, and oxy-fuel carbon capture technologies have gained considerable attention for their potential to curb greenhouse gas emissions from industrial processes and power generation. This economic analysis delves into the cost-effectiveness, efficiency, and viability of these three distinct carbon capture methods. By comprehending the economic landscape of these technologies, policymakers and industries can make informed decisions to drive forward the transition towards a more sustainable and environmentally responsible future [61].

In order to assess the economic performance of the various carbon capture technologies, certain parameters have been identified. Spek et al. stated that, in their economic analysis, they first estimated the capital costs of the suggested system, and then they used the capital costs, in addition to fuel and maintenance costs, to calculate the Levelized Cost of Energy

(LCOE) [61]. The following equations were suggested by Rubin et al. (2013) and have been used [94].

$$LCOE\left(\frac{\text{¢}}{MWh}\right) = \frac{\sum_{i=1}^{n} \frac{I_i + O\&M_i}{(1+r)^i}}{\sum_{i=1}^{n} \frac{E_i}{(1+r)^i}}$$

where I is the investment in a certain year, O&M stands for operation and maintenance cost, r is the discount rate, and E is the electricity production cost in MWh. Furthermore, in order to find the LCOE in the present euro, the following equation is applied by finding the cost of $CO_2$ avoided [95].

$$CCA\left(\frac{\text{¢}}{t\,CO_2}\right) = \frac{LCOE_{CC} - LCOE_{ref}}{C_{ref} - C_{cc}}$$

This equation demonstrates the cost savings achieved by implementing CCS technology in contrast to a scenario without its application, denoted as "CC" for carbon capture and "ref" for the base case without carbon capture technology. It quantifies the financial advantages of employing CCS technology compared to not using it.

However, prior to computing the LCOE, specific financial parameters must be estimated in advance. The capital cost is consistently determined using the process step scoring method, originally introduced by Taylor in 1977 [95]. This method was devised to meet the growing demand for a rapid and effective approach to assessing capital costs for comparison in research and development applications. Despite requiring minimal information, the process step scoring method yields a remarkable accuracy of approximately 95% confidence limits, making it well-suited for preliminary screening in research endeavors.

After calculating the capital cost using Taylor's step method, the total capital cost of the plant can be determined by incorporating the specific CCS technology that has been implemented. As highlighted by Kheirinik et al., the annual release of carbon dioxide from pre-combustion CCS is the highest among the various CCS techniques [95].

Since Taylor's formula for economic calculations dates to 1977, adjusting for the cost index of the current year allows us to find the capital cost for 2023. The index formula is presented below [95]:

$$Capital\,cost_{2023} = \frac{Capital_{1977}}{\frac{Index_{1977}}{Index_{2023}}}$$

By applying the relevant cost index to Taylor's method, we can derive an updated capital cost estimate, reflecting the economic conditions of the year 2023. This approach ensures greater accuracy and relevance in financial projections for the implementation of carbon capture technologies [96].

Table 8 delineates the data calculated for the three CCS technologies. The table includes a comparison of the plants with and without CCS implementation, as well as the costs in 1977 compared to the current year, 2023. The index cost has been obtained from tables published by [97].

**Table 8.** Capital costs illustrated for different CCS technologies with and without CCS implementation.

| Year | Ref | Plant with Pre-Combustion | Plant without Pre-Combustion | Plant Post-Combustion | Plant without Post-Combustion | Oxy-Fuel | Plant without Oxy-Fuel |
|------|-----|---------------------------|------------------------------|-----------------------|-------------------------------|----------|------------------------|
| 1977 | [95] | GBP 16,412,256.00 | GBP 14,706,549.00 | GBP 15,795,650.00 | GBP 11,719,613.00 | GBP 10,277,226.00 | GBP 5,398,367.00 |
| 2023 | Data by the authors | GBP 133,913,187.69 | GBP 119,995,743.21 | GBP 128,882,089.29 | GBP 95,624,314.86 | GBP 83,855,387.97 | GBP 44,047,115.36 |

The economic assessment of the different processes involves multiple cost criteria to consider. For example, the total investment costs include direct and indirect costs, working capital, and on-site capital costs (battery).

Figure 11 presents a graphical representation of the diverse expenses involved in conducting an economic evaluation. These costs encompass fuel expenses, maintenance

and operation costs, construction costs, and more. Additionally, the graph provides an overview of the various costs associated with different carbon capture technologies [98].

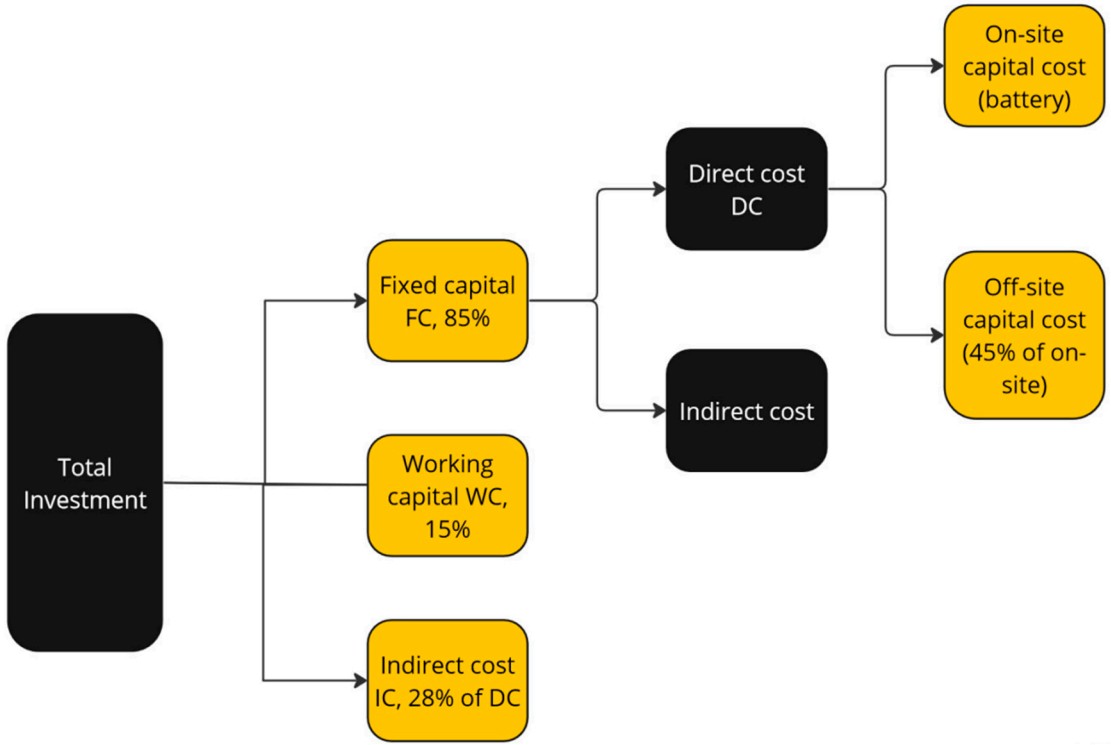

**Figure 11.** Cost categories and sub-categories. Drawn by author.

Figure 12 demonstrates the components considered in calculating the capital cost. Furthermore, Figure 12 highlights that, in comparison to other CCS technologies, the pre-combustion type of CCS incurs the highest expenses.

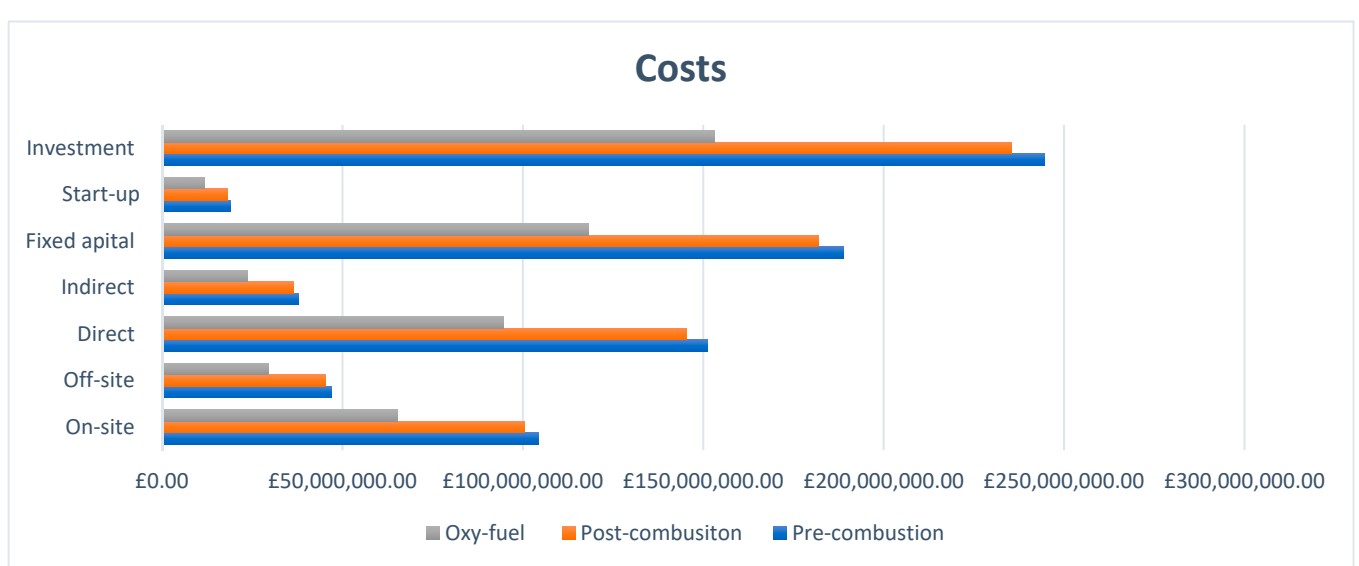

**Figure 12.** Graphical representation of capital cost categories. Data source: [28].

Figure 13 illustrates the differences in investment when using a CCS or without using a CCS.

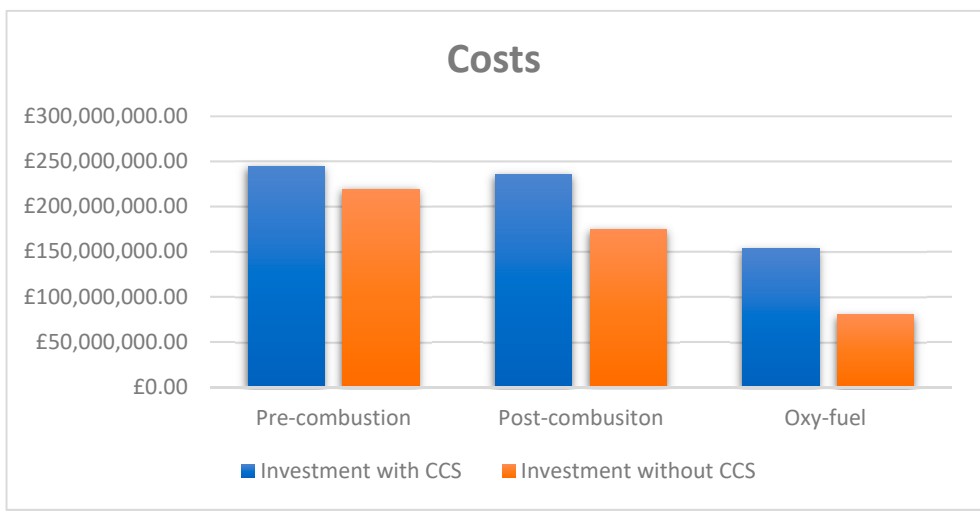

**Figure 13.** Total investment comparison with and without CCS. Data source: [28].

LCOE Analysis

LCOE is a crucial parameter whenever economic analysis is required. The equation mentioned above is used to estimate the LCOE [98]. LCOE values are standardized, and they might vary from one plant to another, but are enough to give an estimate of the economic performance of a certain plant.

Table 9 highlights the important values obtained from the literature when comparing.

**Table 9.** Representative countries $CO_2$ emissions reduction targets.

| Technology | Calculated (GBP/MWh) |
|---|---|
| Post-combustion | 142 |
| PC (without CCS) | 107 |
| Pre-combustion | 148 |
| IGCC (without CCS) | 134 |
| Oxy-fuel combustion | 95 |
| Oxy-fuel (without CCS) | 53 |

Data source: [61].

These values are mere approximations, and they could vary from one location to another due to different discount rates, operation times, and differences associated with costs such as fuel costs from one day to another, O&M costs, etc. Whereas the oxy-fuel plant could differ from one location to another due to the inclusion of the air separation unit (ASU).

## 5. Discussions

This study discussed the advancements in a wide range of CCS technologies and their technical implementation. A critical analysis of the recent publications relevant to the objective of the study was reviewed. From the systematic literature review, certain gaps have been identified by using the VOS viewer software v1.6.18 suite. Keywords were extracted from an online database, and an analysis was conducted on the most commonly discussed areas of research in relation to CCS. In this section, the knowledge gap was identified and explained by using the available data, and visual maps were generated from the VOS viewer software v1.6.18 suite. Future directions are suggested in Sections 5.1 and 5.2 after analyzing original research papers, review articles, and policy reports.

### 5.1. Significance of Carbon Capture Technologies

The analysis using VOS viewer diagrams indicated that research in the field of CCS frequently discusses carbon dioxide, carbon capture, and other variables. However, a

techno-economic analysis of this field in relation to sustainability has not been performed before, which makes this research a crucial piece in the development of carbon capture technologies. Constructing a set of rules and regulations helps assess the techno-economic and social effects of a certain technology. Whereas, from a technical and economic perspective, CCS helps capture $CO_2$ emissions and eliminate possible carbon taxes that may be imposed on countries depending on their carbon release level. That is why policy analysis is crucial in this field to create a suitable ground for the commercialization and implementation of CCS technologies for the industry.

Energy security involves the discussion of a continuous supply of energy, accessing renewable carbon sources at affordable prices, and frequent discussion about decreasing the negative impacts of economic, technical, and environmental threats in energy. It must be noted that energy security is related to a country's national security, which is why discussion of CCS in terms of energy security is crucial.

Feasibility studies deal with the evaluation and analysis of a certain area from the perspective of potential implementation. This study found that there is a knowledge gap in this area, and researching this field is crucial. The implementation of CCS depends on many factors, such as technical performance, economic costs, environmental factors, policies, and public acceptance of the technology. According to the analysis obtained from VOS viewer, it was found that there is no link between CCS and feasibility studies, which indicates a knowledge gap in this area. To implement CCS successfully, a detailed feasibility study should be conducted in terms of economic analysis, geographical location of plants, relevant policies, and system performance, which directly contribute to the development and implementation of carbon capture technologies.

There has been increasing interest in using renewable sources as an energy supplier for CCS. This option is not only feasible economically due to its multiple environmental advantages; it is also considered an important factor in fulfilling the net-zero carbon goal of 2050, as declared by the UN meeting [99]. The pilot study has been conducted in the field of implementing CCS with solar energy and biomass, but a weak correlation was found. However, there is a knowledge gap in the literature when it comes to implementation for wind, geothermal energy, and hydropower. Future studies in CCS technologies could discuss these factors and suggest the feasibility of those sources and their implementation.

CCS technologies require high energy consumption when it comes to storing and extracting $CO_2$, which is why combining CCS with a renewable energy source has proven to be feasible [100]. For example, a conventional method has been applied to combine and capture carbon using solar energy. This method provides a pathway to generate a feasible CCS method. Multiple methods can be used to convert $CO_2$ to renewable products; these include catalytic, electrochemical, mineralization, biological, photocatalytic, and biological processes [53]. The method of using solar energy to convert carbon has gained momentum in the past years. The process entails producing nano-photocatalytic materials and investigating their reaction mechanisms on a laboratory scale. However, the conversion rate of $CO_2$ is currently quite sluggish, often making it unfeasible for commercialization purposes.

The study reveals a significant correlation between carbon dioxide and renewable energy. The progress of CCS technologies relies on multiple factors [27]. To enhance and advance CCS, the decision-making process should incorporate viable assessment criteria to address the complexities of the technology. By examining the interplay between decision-making, energy policies, investments, and uncertainty assessments, the development of CCS can be fortified and expedited. This evaluation involves specific principles, methods, and techniques for the pragmatic appraisal of CCS technologies.

Life cycle assessment stands as one of the evaluation approaches utilized to gauge the environmental implications of CCS technology [8]. An LCA is usually combined with a techno-economic assessment. It consists of the assessment of the technology by considering technical and economic constraints [9]. A lack of studies has been conducted to highlight the significance of the rapid commercialization of these technologies around the globe [16].

A literature gap in this field is crucial for the development of reliable tools to reach the 2050 sustainable development goals.

A comparative study in the field of CCS is crucial for industry operators to decide which technology to choose among the different CCS technologies [101]. According to the analysis, it was found that there is a knowledge gap in the field of comparative analysis of CCS technologies. This knowledge gap has led to an understanding of the correlations between the different CCS types and the importance of different solvents, membranes, solvents, sorbents, etc. [43]. Additionally, comparative studies have been conducted to evaluate two or more potential technologies and identify the best technology to be implemented [41]. The analysis has been conducted within the availability of the most efficient technology, the most suitable to be used in a specific industry, and then after various technologies are accepted [7].

Analyzing patents is an important step in reviewing CCS technologies. It provides an indication of the direction and situation of this technology [19]. In order to obtain a relationship between patents given to CCS technologies and articles released, "Google Patent" database was used to obtain relevant sources. The number of released articles has increased, but in contrast, the number of given patents has steadily decreased in this field. One of the main reasons for this sharp decline could be related to the saturation of the market for this technology or the lack of newly developed technologies in this field. There is a great difference between the patents and the articles published between 2008 and 2013. There is a peaked increase in the emergence of CCS technologies during this time. However, this number has decreased significantly to almost 100 given patents in 2020, as shown in Figure 14.

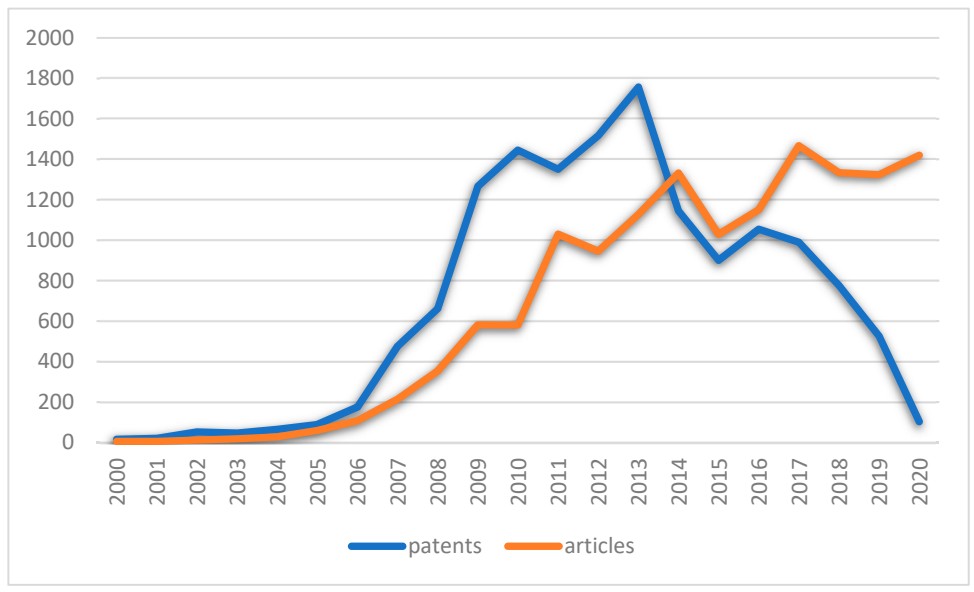

**Figure 14.** Given patents vs. released articles per year. Drawn by author.

In addition to analyzing the number of patents released from the Google Patents database, investigating the companies and certain inventors who obtained those patents could reveal reliable information about the situation of CCS. According to Khosroabadi et al. (2021), the most important part is that most of the patent owners were companies, research centers, and inventors [101]. According to the study, Tianjin University emerged as a significant contributor with the highest share of patents at 3.3%, followed by General Electric Company in second place with a 2.5% share (ibid). Alstom technology in China secured the third position with a share of 1.7%, and ExxonMobil Research and Engineering, operating in the US, ranked fourth, with a share of 1.5%.

Additionally, the Korea Institute of Energy Research holds a 1.5% share of the patents [82]. Consequently, China, the USA, and Korea collectively possess the majority of patents,

underscoring their pivotal roles in the advancement of CCS technologies. Several studies have revealed that most of the patents have been submitted by companies. This indicates that CCS is on the way to commercialization. This also explains the high number of patents in contrast to the number of released articles. Companies investing in CCS and owning patents have great potential to shape the future of CCS technologies. This paves the way for the rapid implementation and commercialization of the different CCS technologies. Furthermore, most of the inventors in this field are Chinese companies.

*5.2. Risk Analysis of Different CCS Technologies*

The results from the different analyses were conducted in an economic study. CDR implementation with other technologies and the feasibility of the study are shown in Table 8. The findings highlighted the effects of different scenarios when CDR is implemented and when it is not implemented in the future. As can be seen in Table 8, implementing carbon removal technologies influences long-term $CO_2$ emissions. The next question that should be discussed is the cost of the implementation of these technologies. Table 10 shows the technology and the cost of some uncertain technologies.

**Table 10.** Technologies with uncertain parameters.

| Technology | Techno-Economic Parameter | Ranges (2050) | | |
|---|---|---|---|---|
| | | **Low** | **Medium** | **High** |
| Wind | Capital cost USD 2005/kW | 4800 | 830 | 1290 |
| Solar | Capital cost USD 2005/kW | 180/1500 | 230/2350 | 420/3300 |
| Nuclear | Capital cost USD 2005/kW | 2700 | 6000 | 8750 |

Data Source: [9].

To understand the impact of technology on various mitigation factors, it is necessary to conduct multiple analyses. This process begins by identifying and quantifying the sources of uncertainty to estimate the effect of technology costs on different mitigation factors. To draw a conclusion, a series of sensitivity analyses may also be required. The mitigation factors can be organized into three thematic groups to provide a clearer understanding of the effect of technology costs. Table 11 shows the mitigation indicators and their corresponding groups.

**Table 11.** Model output characteristics of CDR and YCN mitigation indicators.

| | Group | Metric | Sector | Year | Aggregated |
|---|---|---|---|---|---|
| 1. | Economics | Carbon tax | No | 2100 | No |
| | | Climate policy costs | No | 2010–2100 | |
| 2. | Depth | $CO_2$ emissions | Yes | 2030–2050 mean | No |
| | | Electricity share of final energy | Yes | 2030–2050 mean | No |
| | | Fossil carbon intensity of fuels | Yes | 2030–2050 mean | No |
| 3. | CDR and YCN | Aggregated CDR | No | 2020–2100 | Yes |
| | | Year Of Carbon Neutrality | No | - | No |

Data source: [9].

Gardarsdottir et al. (2019) conducted a sensitivity analysis to examine the various impacts of technologies on costs [74]. Overall, the implementation of $CO_2$ removal (CDR) technologies added flexibility to the system, resulting in a stronger impact of energy technologies on mitigation compared to cases where CDR was not used. The carbon capture and sequestration rate had the greatest impact, except in the case of unconstrained injection, where CDR was reduced but still had a significant impact, as indicated by the blue line, which was much smaller compared to the low injection of CDR, which was reduced by 50%.

### 5.3. Limitations of CO$_2$ Capture Capacity

While discussing the economic viability of carbon capture and utilization technologies, a crucial point that should be addressed is the limits at which capturing carbon is possible. According to Mikulčić et al. (2019), the standard technique used for developing a comparison pattern is by using the physical adsorption of CO$_2$ at 273 K and nitrogen at 77 K to assess the maximum absorption capacity in both post- and pre-combustion conditions [102]. Using the adsorption isotherm and comparing the theoretical relevance of Dubinin's theory leads to an estimation of the equilibrium CO$_2$ at different levels. Experiments revealed that a CO$_2$ absorption of around 10–11% seems to be realistic for standard activated carbons under post-combustion.

The International Energy Agency (IEA) developed a series of roadmaps to achieve the wide commercialization of carbon dioxide technologies [26]. However, one of the roadblocks standing in the way of achieving this dream is that the technical capability of the technology as well as the associated cost of production need to be lowered further in order to reach worldwide commercialization [53]. After testing multiple post- and pre-combustion technologies, it was realized that absorption presents the highest adsorption capacity, great selectivity, good mechanical properties, and stability over repeated adsorption–desorption cycles [22]. Some materials that are commonly used commercially are zeolite, alumina, silica gel, and activated carbon. They are popular for separating carbon dioxide from the gas mixture [13].

Moreover, activated carbon is a good candidate among the mentioned materials; their adsorption capacity is dependent on the pore structure and the structure chemistry of the activated carbon [89]. Whereas their capturing capacity is lower than those of zeolite and molecular sieves under low pressure, they have an advantage at high pressures to absorb carbon dioxide due to their ease of regeneration, low cost of manufacturing, and lower sensitivity to moisture. A number of the carbon absorbents mentioned are sensitive to moisture, in contrast to the behavior of zeolite [1]. The impact of other factors on the behavior of this material would have to be neglected because they would be negligible due to the low absorption capacity of low concentration levels and therefore do not affect carbon absorption.

### 5.4. Life Cycle Analysis for CCU

The life cycle analysis for carbon removal technologies is discussed in this section. First, highlighting the sustainability of carbon removal technologies is an important criterion to consider when mentioning carbon removal technologies. For example, certain adsorbents are being used for the absorption of carbon dioxide, and these chemicals could be toxic, poisonous, and corrosive and regeneration of the solvent would be difficult [89]. In that case, these solvents would not make much sense to use. In fact, certain chemicals and solvents could be difficult to recycle, which would make the whole concept of carbon capture and removal unworthy since we are fixing one part while damaging the other [40].

Analysis of the LCA of these technologies would be concentrated on the chemical solvent as well as the construction life cycle of the process unit, as shown in Figure 15 [4]. Certain software would be used for this analysis. In fact, several researchers have already developed certain guidelines and regulations for carbon removal technologies [51]. These regulations encompass multiple criteria and follow certain guidelines enacted by the Intergovernmental Policy on Climate Change (IPCC). Furthermore, it was realized that applying LCA to CCU leads to the discovery of certain methodological issues, such as the double role of carbon dioxide as an emission and a feedstock [24]. The guidelines were developed with the collaboration of 40 experts in the field of climate change, including researchers, policymakers, and instructors. By following the guide made by López et al. (2023), transparency when it comes to creating a comprehensive guide on the life cycle has been enhanced [103]. The guide used for this process is a cradle-to-grave aspect of energy policy-making decisions.

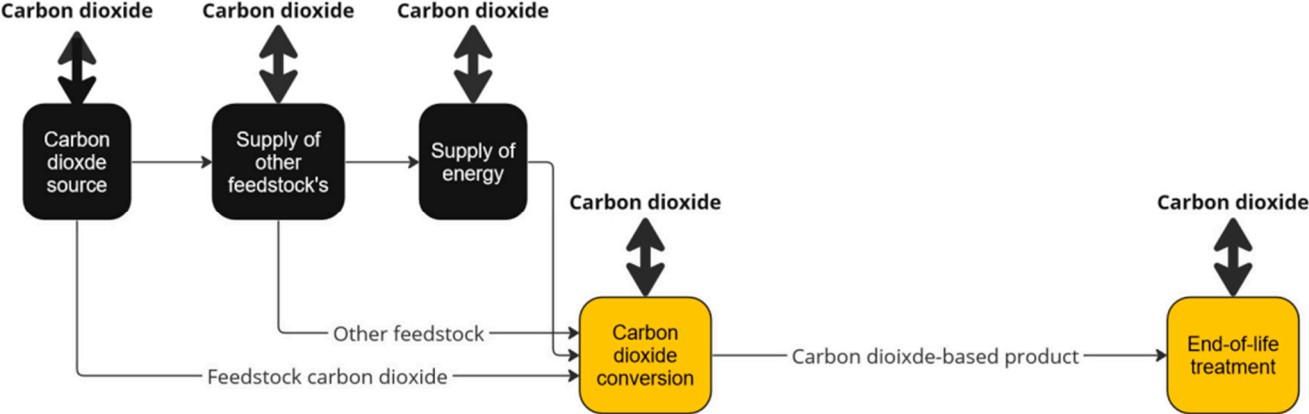

**Figure 15.** Life cycle analysis of a carbon capture and removal process unit. Drawn by author.

*5.5. Government Policies in Connection to CCS*

The publication of the WG3 IPCC report emphasized the need to develop and accelerate the deployment of carbon capture and sequestration technologies. The report emphasized that to lower global temperatures by 1.5 °C, effective carbon capture and storage (CCS) and carbon dioxide removal (CDR) technologies need to be put into action [104]. The IPCC report highlighted carbon capture and storage (CCS) as a crucial technology for mitigating climate change and achieving global emission reduction goals. The report emphasized the following key points about CCS:

-   Effectiveness in Emission Reduction: The IPCC report recognized CCS as an effective method for reducing greenhouse gas emissions, particularly from large-scale fossil-based energy and industrial sources. It acknowledged that CCS could play a significant role in limiting global warming to the desired targets, such as the 1.5 °C warming limit outlined in the Paris Agreement [105].
-   Feasibility and Viability: The report underscored the technical feasibility and viability of CCS technologies. It provided evidence and modeling scenarios that demonstrated the potential for widescale deployment of CCS across various sectors to achieve emissions reductions.
-   Complementary Role with Carbon Dioxide Removal (CDR): The IPCC report highlighted the complementary relationship between CCS and technology-based carbon dioxide removal (CDR) methods. It recognized that while emissions reduction efforts are crucial, there will still be residual emissions that need to be counterbalanced with CDR and CCS to achieve net-zero emissions.
-   Role in Specific Sectors: The report emphasized the importance of CCS in certain sectors, such as heavy industry and power generation. It identified CCS as a key tool for decarbonizing these sectors, which are particularly challenging to address through other means.
-   Integration with Sustainable Development Goals (SDGs): The report analyzed the relationship between CCS and the Sustainable Development Goals (SDGs). It identified synergies between CCS deployment and specific SDGs, demonstrating how CCS can contribute to broader sustainable development objectives.
-   Policy Support and Investment: The IPCC report called for increased policy support and investment in CCS technologies. It encouraged governments and stakeholders to create favorable regulatory frameworks and financial incentives to promote the development and deployment of CCS projects.

Additionally, the IPCC report recognized CCS as a valuable and necessary component of global climate change mitigation efforts [106]. It served as a significant endorsement of CCS technologies and contributed to raising awareness and support for their widespread adoption as part of the solutions to combat climate change [107]. By endorsing policies that encourage carbon capture technologies in power generation, policymakers can expedite

the low-carbon transition [107]. Government policies regarding carbon capture and storage (CCS) vary from country to country, as each nation has its unique energy mix, climate goals, and policy priorities [108]. However, several common policy approaches are often seen across different regions to promote and incentivize CCS deployment. Here are some typical government policies in connection with CCS [109]:

- Financial Incentives and Subsidies: Governments may offer financial incentives, grants, or subsidies to support the development and deployment of CCS technologies. These incentives can help reduce the high upfront costs associated with building and operating CCS facilities and encourage private investment in such projects.
- Carbon Pricing: Many countries implement carbon pricing mechanisms, such as carbon taxes or cap-and-trade systems, to put a price on carbon emissions. By valuing $CO_2$ reductions, these policies create economic incentives for industries to invest in CCS technologies as a means to lower their carbon liabilities and comply with emissions reduction targets.
- Regulatory Requirements and Emission Standards: Governments may establish regulations or emission standards that require certain industries or large emitters to reduce their greenhouse gas emissions. CCS can be considered a compliance option, providing companies with an alternative way to meet their emission reduction obligations.
- Research and Development Funding: Governments often invest in research and development (R&D) programs focused on advancing CCS technologies. These funding initiatives aim to improve the efficiency, cost-effectiveness, and safety of CCS systems, making them more viable for widespread deployment.
- Supportive Policy Frameworks: Governments can create comprehensive policy frameworks that prioritize low-carbon technologies, including CCS, within national energy and climate strategies. This involves setting clear long-term goals and targets for emissions reductions and providing a roadmap for CCS integration into the energy sector.
- Carbon Capture Utilization and Storage (CCUS) Deployment Roadmaps: Governments may develop deployment roadmaps outlining the steps and timelines for the large-scale adoption of CCUS technologies. These roadmaps facilitate coordination among stakeholders and provide a clear vision for the development and deployment of CCS projects.
- Public–Private Partnerships: Governments may establish partnerships with private companies and research institutions to accelerate the development and deployment of CCS technologies. Such collaborations can leverage expertise, resources, and funding to overcome technical and financial challenges.
- International Collaboration and Agreements: Many governments participate in international agreements and initiatives to cooperate on CCS research, development, and deployment. Sharing knowledge and experiences with other countries can help accelerate the global deployment of CCS technologies.
- Support for CCS Infrastructure: Governments may provide support for the development of CCS infrastructure, such as $CO_2$ transport and storage networks, to encourage the growth of a viable CCS industry.

Overall, government policies in connection with CCS play a crucial role in creating an enabling environment for the widespread adoption of these technologies, supporting climate goals, and facilitating the transition to a low-carbon future [110,111].

## 6. Conclusions

This paper reviews carbon capture and sequestration technologies to mitigate the issue of $CO_2$ emissions in the world. Many research projects have been conducted to develop state-of-the-art carbon capture technologies, but investigating the issue of the economic feasibility of those technologies has been neglected. The most important and relevant topic of CCS is post-combustion carbon capture, which consists of capturing carbon using

absorption technologies such as adsorption, membranes, or cryogenics. This review paper discussed both the strengths and weaknesses of current CCS technologies to provide a roadmap during the decision-making process. The study found that further research is required to improve upon them for large-scale usage. One of the important findings is the cost-effectiveness of the technologies and their worldwide applicability at an affordable price. It was found that CCS has great potential to mitigate global carbon concentrations in the atmosphere.

The mixed-methods design was adopted to investigate the various determinant factors that influence the implementation of carbon capture and storage (CCS) in industrial and domestic settings. The study considers two main aspects, technical and economic, to investigate the significance of CCS technologies globally. In the economic part, there are several criteria that affect the cost of CDR technology implementation, such as energy technology costs each year. Additionally, different climate policies, such as the no-policy baseline (BASE), the currently announced Paris pledges (NDC), and the three climate scenarios (B900, B1100, B1300), result in different $CO_2$ emissions and global temperatures, which can affect the economic analysis. The study found that the REMIND model is the most applicable conceptual framework to analyze the uncertain costs of multiple energy changes for different regions. The study also analyzed the impact of different energy suppliers and their costs, as well as the availability of CDR, on mitigation by performing a sensitivity analysis using the REMIND energy-economy climate model. The sensitivity analysis indicated that the uncertainty in biomass and CCS, followed by transport operations, had the greatest effect on the mitigation parameters.

Furthermore, the technical feasibility of the different post-capture carbon capture technologies was discussed. It must be stressed that oxy-fuel combustion (OFC) technology is the most efficient among the post-capture technologies, while post-capture methods were observed to have the highest efficiency and implementation factor compared to pre-capture methods. In fact, the application of OFC in waste incineration plants could result in negative $CO_2$ emissions. Future studies should include further analysis of the different adsorption and absorption technologies and their feasibility of being implemented on a commercial scale. Further enhancement of the analysis sector should be investigated to fill the knowledge gap in carbon capture technologies worldwide.

**Author Contributions:** Conceptualization, R.A. and B.O.; methodology, R.A. and B.O.; software, R.A. and B.O. validation, R.A. and B.O.; formal analysis, R.A.; investigation, R.A.; resources, R.A. and B.O.; data curation, B.O.; writing—original draft preparation, R.A. and B.O.; writing—review and editing, R.A. and B.O.; visualization, R.A.; supervision, B.O.; project administration, B.O. All authors have read and agreed to the published version of the manuscript.

**Funding:** The work presented is the outcome of the SEES 586: Environmental design and engineering course's research output as part of the MSc in Sustainable Environment and Energy Systems (SEES) Graduate Program at the Middle East Technical University Northern Cyprus Campus.

**Institutional Review Board Statement:** Not applicable.

**Informed Consent Statement:** Not applicable.

**Data Availability Statement:** Data will be made available upon request.

**Acknowledgments:** The authors would like to acknowledge the Sustainable Environment and Energy Systems (SEES) Graduate Program at the Middle East Technical University Northern Cyprus Campus.

**Conflicts of Interest:** The authors declare no known competing financial interests or personal relationships that could have appeared to influence the work reported in this paper.

## Abbreviations

| | |
|---|---|
| COP21 | 21st Conference of The Parties |
| GHG | Greenhouse gases |
| IPCC | Intergovernmental Panel on Climate Change |
| UNFCC | United Nations Framework Convention on Climate Change |
| CCS | Carbon capture and storage |
| CDR | Carbon dioxide removal |
| LCA | Life cycle assessment |
| ASU | Air separation unit |
| IGCC | integra |
| OFC | Oxy-fuel combustion |
| NCS | Natural climate solutions |
| BECCS | Bioenergy With carbon capture and storage |
| DACS | Direct air capture and storage |
| OFC | Oxy-fuel combustion |
| CHP | Combined heat and power |
| LCOE | Levelized cost of energy |
| NPV | Net present value |
| PV | Present value |
| SBC | Sodium bicarbonate slurry |
| CSP | Concentrated solar power |
| TGA | Thermogravimetric analysis |
| FTIR | Fourier transform infrared |
| GC | Gas chromatography |
| MS | Mass spectrometry |
| CD | Carbon emissions |
| GPD | Economic development |
| P | Population |
| E | Energy production |
| C | Carbon-based fuel |
| $S_{CO_2}$ | Carbon dioxide sinks |
| CFD | Computational fluid dynamics |
| FOQUS | Framework for optimization, quantification of uncertainty and surrogates |
| COPLOS | Communication about prospects and limitations of simulation results for policymakers |
| PIMS | Polymers of intrinsic microporosity |
| YCN | Year of carbon neutrality |

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
