# Peer review of "Techno-Economic Analysis of State-of-the-Art Carbon Capture Technologies and Their Applications: Scient Metric Review"

_encyclopedia, doi:10.3390/encyclopedia3040092_

Round 1

Reviewer 1 Report

The manuscript reports EA Analysis for CO2 capture technologies. Main conclusoin about post-combustion is correct. Suggestions for modifications:

1. Besides appeared in topic, abstract, references and section 'Introduction', the word 'building' is not appeared in the body of the manuscript. If there are no detailed discussion about CO2 capture related to 'building', it is necessary to delete 'building' in the whole manuscript.

2. For EA analysis of CO2 capture technologies,  oxy-fuel combustion could not be the prefered one because of high cost of ASU. Please refer to more literature about it, and make more sound conclusion about this.

Author Response

We sincerely thank you for accepting our research work with revisions. In this revised version, we followed both reviewers’ recommendations and re-conceptualised the entire paper to well fit in the scope of the Encyclopaedia.

Besides revisions, to increase the readability of the manuscript, we decided to revise the manuscript substantially in order to increase scientific soundness and credibility of this research work.

For your information, this research work was set out to inform policy makers on the development of the state-of-the-art carbon capture technologies (CCT). We also addressed the economic analysis of the CCT, which has not been addressed by previous scholars at the time of developing policy design for the industry. This is the main research gap addressed in this empirical study and we set out an extensive presentation of our work to be published in the Encyclopaedia. We outlined each reviewers’ recommendations as follows:

A.1 Besides appeared in topic, abstract, references and section 'Introduction', the word 'building' is not appeared in the body of the manuscript. If there are no detailed discussion about CO2 capture related to 'building', it is necessary to delete 'building' in the whole manuscript.

R.1 We thank for the reviewer about this correction. We removed the building from the title of the paper. We also removed the terminology of the ‘building’ throughout the manuscript where it is necessary. We agree on the point that the ‘building scale’ is not within the scope of our work. In this systematic review paper, we are aimed to demonstrate the effectiveness of carbon capture technologies for any type of industry globally. We sincerely thank to reviewer about this correction once again.

A.2 For EA analysis of CO2 capture technologies, oxy-fuel combustion could not be the preferred one because of high cost of ASU. Please refer to more literature about it, and make more sound conclusion about this.

R.2 We thank for reviewer about highlighting the importance of economic analysis to prove wider applicability of the carbon capture technologies (CCT). In the revised version, we added new sub-section – 4.5 (Economic Analysis of CCS Technologies) see pages between 25-28, lines between 858-946. In this newly added section, we obtained data from the open-access secondary data source, and we conducted our economic analysis by referring with the current literature in the CCT. First, we conducted critical review on the economic appraisal of the CCT. Second, we included Tables 8 and 9 by calculating the data extracted from the secondary data source and we conceptualised Figures 14 and 15 to demonstrate the economic analysis of these technologies. Additionally, we included the Figure 13 to tabulate the cost categories clearly.

Reviewer 2 Report

In this review article, the authors have reported the state-of-the-art carbon capture and storage (CCS) technologies and their implementation of an evidence-based energy policy design. This study reviews the current methods of CCS in three phases  (a) post-combustion (b) pre-combustion and (c) oxy-fuel combustion processes.

1. The approach seems good but most of the new carbon capture techniques like the use of biomass for CC are not included in the description.

2. Carbon storage mechanisms such as biological carbon sequestration, and geological carbon sequestration need to be added.

3. In addition to this, the article needs to explain more in detail about the storage of carbon and its possible applications. 

4. Need to add some descriptions for the possible causes of carbon in the environment and its harmful effect on the society and environment.

5. The technology with diagrammatic representation for carbon capture and storage needs to add to the manuscript.

6. Most of the CCS techniques were explained only on a single model using the VOS viewer software suite. More explanation is required for other CCS models.

7. Most preferably the authors need to explain various techniques for CC and CS adopted using recent technological advances.

8. Government policies in connection with CCS need to be added to the manuscript.

9. No pictorial representation of adsorption technologies in CCS. You can ask permission from the corresponding authors of the manuscripts before the picture citation.

10 Most importantly, the authors need to explain more about CC and its storage, however, the present manuscript has mostly explained oxyfuel combustion, and seems it is diverted from the track.

More careful refinement is required to bring the manuscript to acceptance level.

Good luck...!

Needs improvement.

Author Response

We sincerely thank you for accepting our research work with revisions. In this revised version, we followed both reviewers’ recommendations and re-conceptualised the entire paper to well fit in the scope of the Encyclopaedia.

Besides revisions, to increase the readability of the manuscript, we decided to revise the manuscript substantially in order to increase scientific soundness and credibility of this research work.

For your information, this research work was set out to inform policy makers on the development of the state-of-the-art carbon capture technologies (CCT). We also addressed the economic analysis of the CCT, which has not been addressed by previous scholars at the time of developing policy design for the industry. This is the main research gap addressed in this empirical study and we set out an extensive presentation of our work to be published in the Encyclopaedia. We outlined each reviewers’ recommendations as follows:

A.1 The approach seems good but most of the new carbon capture techniques like the use of biomass for CC are not included in the description.

R.1 We thank the reviewer for reminding us about the objective of this study. To prove the intended objective, which was initially set up by the research consortium, we added more information in Introduction to outline the significance of carbon capture technologies (CCT), see page 4, lines between 134-145. To provide consistency and meaningful narrative structure to the paper, we added new-subsection – 2.1 (Causes and Effects of Carbon Dioxide in the Atmosphere) and we provided the necessary description within the first paragraph.

A.2 Carbon storage mechanisms such as biological carbon sequestration, and geological carbon sequestration need to be added.

R.2 In the Literature review section, we added new sub-section 2.8 (Carbon Sequestration and Utilisation), see pages between 14-16, lines between 459-528. We also added heading entitled ‘Geological Sequestration & Storage), in this section, we discussed the principles of implementing the technologies in order to provide an effective roadmap to the policymakers.

A.3 In addition to this, the article needs to explain more in detail about the storage of carbon and its possible applications. 

R.3 In the Literature review section, we added new sub-section, Industry Utilisation, see pages between 16-17, lines between 530-612. We continued to discuss the significance of the carbon sequestration and utilisation outlined previously included into the new subsection.

A.4 Need to add some descriptions for the possible causes of carbon in the environment and its harmful effect on the society and environment.

R.4 In the Literature review section, we added new sub-section 2.1 (Causes and Effects of Carbon Dioxide in the Atmosphere), see pages between 5-5, lines between 135-188. In this section, we discuss the carbon capture technologies and its impact on the society and environment. We thank to the reviewer for his/her helpful insight. We agree on the point that it is an important to highlight the environmental impact of the state-of-the-art technologies in this review paper.

A.5 The technology with diagrammatic representation for carbon capture and storage needs to add to the manuscript.

R.5 We designed a flow-diagram to demonstrate the adsorption process applied to carbon capture technologies – see Figure 7 at page 14.

A.6 Most of the CCS techniques were explained only on a single model using the VOS viewer software suite. More explanation is required for other CCS models.

R.6  In this paper, Vos viewer software suite was used to visualise the keywords selected within the scope of the study – see Figure 4, at page 11. Additionally, we included descriptive information to outline the chosen method for the CCS models, see at page 21, lines between 733-738.

A.7 Most preferably the authors need to explain various techniques for CC and CS adopted using recent technological advances.

R.7 We opened up a new sub-section 2.7 (Carbon Dioxide Separation Technologies), see pages between 13-14, lines between 423-457. We also added a new flow-diagram – see the Figure 7.

A.8 Government policies in connection with CCS need to be added to the manuscript.

R.8 We thank to the reviewer about this recommendation. To increase the credibility and scientific soundness of the paper, we added new sub-section 5.5 (Government Policies in Connection to CCS), see pages between 36-38, lines between 1173-1280. For the reviewer’s information, most-up-to-date government policies were reviewed and critically analysed within the scope of this manuscript.

A.9 No pictorial representation of adsorption technologies in CCS. You can ask permission from the corresponding authors of the manuscripts before the picture citation.

R.9 

We designed a flow diagram see the Figure 7, at page 14. Additionally, we confirm that the Figures between 10-12 and 16-22 were obtained from the open-access paper. Here is the link for your reference – https://www.sciencedirect.com/science/article/pii/S2666790821002391?via%3Dihub

As you are acknowledged that open-access papers do not require any copyright permission but we informed the authors of the paper and obtained the copyright permission certificate from the publisher. We also informed the responsible associate editor who is handling with our paper and the editor has obtained the necessary permissions for those indicated images. We sincerely thank for this important ethical information about those images. We really appreciate for your guidance on this matter.

A.10 Most importantly, the authors need to explain more about CC and its storage, however, the present manuscript has mostly explained oxyfuel combustion, and seems it is diverted from the track.

R.10 In this revised version of the manuscript, we added new sub-section – 2.8 (Carbon Sequestration and Utilisation), see pages between 14-17, lines between 459-612. In this section, we extensively discussed the significance of the carbon sequestration process and its implications. In order to provide consistent critical review within the scope of the paper, we also added new sub-section – 4.5 (Economic Analysis of CCS Technologies), see pages between 25-28, lines between 858-946. We added new information to comply with the scope of the paper. We sincerely thank to the reviewer about his/her recommendation to avoid any confusion to the future readers.

Round 2

Reviewer 1 Report

The revised version has included modifications according to comments. And I suggest to publish it as it is now.

Reviewer 2 Report

Accept 

Needs moderate corrections